# Low complexity RGG-motif sequence is required for Processing body (P-body) disassembly

Raju Roy [1], Gitartha Das[1], Ishwarya Achappa Kuttanda[1], Nupur Bhatter[1] & Purusharth I. Rajyaguru [1✉]

P-bodies are conserved mRNP complexes that are implicated in determining mRNA fate by affecting translation and mRNA decay. In this report, we identify RGG-motif containing translation repressor protein Sbp1 as a disassembly factor of P-bodies since disassembly of P-bodies is defective in Δsbp1. RGG-motif is necessary and sufficient to rescue the PB disassembly defect in Δsbp1. Binding studies using purified proteins revealed that Sbp1 physically interacts with Edc3 and Sbp1-Edc3 interaction competes with Edc3-Edc3 interaction. Purified Edc3 forms assemblies, promoted by the presence of RNA and NADH and the addition of purified Sbp1, but not the RGG-deletion mutant, leads to significantly decreased Edc3 assemblies. We further note that the aggregates of human EWSR1 protein, implicated in neurodegeneration, are more persistent in the absence of Sbp1 and overexpression of EWSR1 in Δsbp1 leads to a growth defect. Taken together, our observations suggest a role of Sbp1 in disassembly, which could apply to disease-relevant heterologous protein-aggregates.

---

[1] Department of Biochemistry, Indian Institute of Science, Bangalore 560012, India. ✉email: rajyaguru@iisc.ac.in

RNA granules are conserved, membraneless dynamic condensates consisting of RNA-protein complexes. They play an important role in the regulation of mRNA translation and decay to modulate cell proteome in response to specific physiological cues. P-bodies (PBs) and stress granules (SGs) are the most well-characterized forms of RNA granules. SGs and PBs share many protein components and exchange mRNPs[1–3]. SGs harbor mRNAs in translationally repressed forms in complex with various translation factors[4–6]. On the other hand, PBs also contains enzymes and proteins that promote mRNA decay, leading to the degradation of resident mRNAs[7,8]. Interestingly in yeast, SGs arise from pre-formed PBs under glucose deprivation stress indicating that PBs acts as repositories of mRNPs that are either degraded or subsequently transferred to SGs[6].

PBs exist at a basal level under unstressed conditions in some cell types, including yeast, and are strongly induced in response to several stresses such as glucose starvation, oxidative stress, and heat stress which lead to global translation repression[6]. A recent report indicates that PBs are important for maintaining stem cell plasticity as the loss of PBs lead to a differentiation-resistant state of the embryonic stem cells[9]. Some PB-resident proteins regulate mRNAs involved in inflammatory response[10]. Virus infections, specifically by RNA viruses, lead to the disassembly of PBs, indicating that PBs could play a role in viral stress response[11]. PBs have been reported to contain mRNA encoding specific proteins involved in various cellular processes. Analysis of PB-resident mRNAs provides a striking observation. Transcripts encoding regulators of important processes such as protein turnover, RNA processing, cell cycle, and energy metabolism are compartmentalized to PBs. However, the transcripts that encode proteins with constitutive and structural functions are excluded[12]. Such arrangements highlight the role of PBs in specifically regulating the regulators. Therefore, understanding PB assembly and disassembly would be crucial to elucidating regulation of important cellular processes.

The mechanistic basis of PB assembly has been addressed. An important factor that modulates RNA granule formation, in general, is Intrinsically Disordered Regions (IDRs) in RNA granule associated proteins[13]. IDRs often contain repeats of specific amino acid residues (Low complexity sequences) that cannot form a three-dimensional structure but can be involved in binding to proteins and/or nucleic acids to contribute towards specific functions[14]. Some examples of Low complexity (LC) sequences include repeats of RGG/RG (Arg-Gly-Gly), YGG (Tyr-Gly-Gly), QQQ (or, polyQ), GY/GSYGS/ GYS/SYG/ SYS, Q/N (Gln-Asn), RS (Arg-Ser) and PPP (Proline-rich motif). These LC region containing proteins are known to bind to other proteins and, in specific cases, can bind to self, leading to the formation of higher-order structures[15]. Edc3 is a conserved marker of PBs and plays a vital role in their assembly. The C-terminal YjeF-N domain of Edc3 protein binds itself to promote the formation of higher-order PB assemblies[16]. Deletion of Edc3 reduces the ability of cells to form PBs. However, Δedc3 cells are highly defective in PB assembly in the absence of the C-terminal Q/N-rich prion-like motif of PB protein Lsm4[16]. Consistent with the idea of Edc3 self-association in yeast, other PB proteins are known to multimerize and contribute to PB assembly. Human EDC3 forms dimers[17], fly DCP1 forms trimers[18], and human RCK/p54 can form multimers[19]. Similarly, PGL-3, a key germ granule assembly factor in *C. elegans*, self-associates and binds other RNA and mRNP components through its RGG-motif[20]. TIA-1, an essential stress granule protein in humans, facilitates SG assembly via aggregation of the Q/N-rich region[21,22].

The LC sequences participate in multiple low-affinity interactions to undergo Liquid–Liquid Phase Separation (LLPS) that promote RNA granule formation[23,24]. RNA often provides a scaffold for such interactions between the RNA-binding proteins with LC sequences to augment LLPS[25]. A strong link has emerged between aberrant phase transitions of RNA granule components and neurodegenerative disorders. FUS and TDP43 (RGG-motif containing nuclear proteins involved in Amyotrophic Lateral Sclerosis) are reported to accumulate mutations that lead to the formation of cytoplasmic amyloid inclusions which are linked to the disease[26,27]. Similar observations are also reported for other nuclear RBPs such as hnRNPA1, hnRNPA2 and EWSR1[28,29]. These examples highlight that persistent aberrant RNA granules that fail to disassemble play an important role in these neurodegenerative disorders[30]. Therefore, understanding the factors contributing to disassembly is paramount as this information could help devise strategies to deal with disassembly-resistant aberrant RNA granules. In this study, we provide evidence that a low-complexity RGG-motif sequence is important for PB disassembly. Deletion of Sbp1, an RGG-motif containing translation repressor and granule resident protein, renders disassembly of PBs defective. We further report that purified Edc3 forms phase-separated assemblies in vitro promoted by RNA and NADH. Strikingly, purified Sbp1 leads to a significant decrease in the preformed Edc3 assemblies in a manner that is dependent on its RGG-motif.

## Results

**Δsbp1 is defective in Edc3 granule disassembly.** RGG-motif proteins have been implicated in the assembly of RNA granules[31–33]. However, the role of low complexity sequences like RGG-motif in disassembly has not been explored. Therefore, we addressed the role of RGG-motif proteins Scd6 and Sbp1 in RNA granule disassembly. To test the hypothesis, we took wild-type strains of *Saccharomyces cerevisiae* (BY4741) and strains where the known translation repressors, Scd6[31,32] and Sbp1[34,35] were deleted, respectively. A *CEN* plasmid co-expressing Pab1-GFP (a stress granule marker)[36] and Edc3-mCherry (a P-body marker)[36] was transformed into wild-type, Δscd6 and Δsbp1 strains, respectively. Wild-type strain upon sodium azide stress showed induction of stress granules (SG) and P-body (PB) formation consistent with earlier reports[36]. These granules disassembled upon removal of stress during recovery (Fig. 1A, wild-type microscopy panel; Fig. 1B). Δscd6 strain showed reduced stress granule formation, as reported earlier[31], but P-body assembly was comparable to the wild-type strain (Fig. 1A, Δscd6 microscopy panel; Fig. 1B). Δsbp1 strain also showed defect in stress granule formation but normal P-body formation as compared to wild-type strain (Fig. 1A, Δsbp1 microscopy panel, +sodium azide, Fig. 1B). A striking observation was that Δsbp1 was defective in P-body disassembly even after 1 h of recovery (Fig. 1A, Δsbp1 microscopy panel, recovery condition; Fig. 1B). The disassembly defect of Edc3-mCherry in Δsbp1 strain is not due to increased levels of Edc3-mCherry protein during recovery (Fig. 1C). The disassembly of stress granules was found to be normal (Fig. 1A, Δsbp1 panel, recovery condition). A similar disassembly defect was observed for Edc3 foci formed from endogenously tagged Edc3-mCherry strain in Δsbp1 background (Fig. 1D, E). Based on these results, we conclude that disassembly of Edc3 foci is defective in Δsbp1 strain.

**Disassembly of other P-body markers is defective in Δsbp1.** We further investigated if Δsbp1 affects disassembly of other P-body components such as Dhh1 and Scd6. Dhh1 is a PB component[37] that is important for P-body assembly. Interestingly Scd6 localizes to both PBs and SGs[31], indicating that it could be a protein that facilitates remodeling of mRNPs from PBs to SGs. We transformed wild-type and Δsbp1 strains with *CEN* plasmid expressing

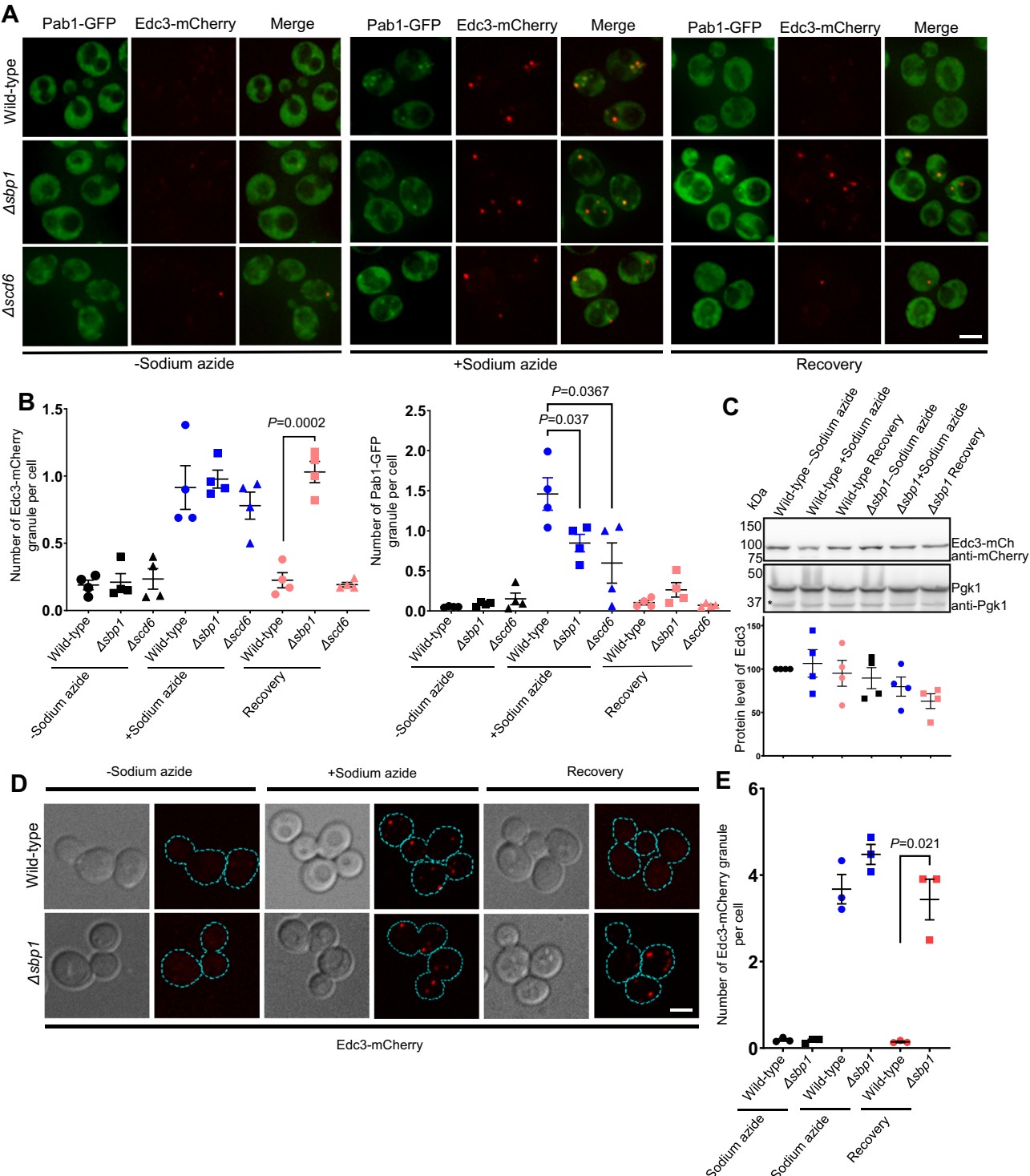

**Fig. 1 Disassembly of Edc3 granules is defective in *Δsbp1*. A** *Δsbp1* leads to Edc3 granule disassembly defect. Cells were cultured till 0.35–0.4 OD$_{600}$ and incubated for 30 min with or without 0.5% (v/v) sodium azide at 30 °C. Subsequently, cells were pelleted and washed thrice with glucose-containing medium. For stress recovery, the resuspended cells were grown for an additional 1 h at 30 °C in media without sodium azide. Scale Bar, 3 μm. **B** Graph depicting the number of foci per cell in wild-type, *Δsbp1* and *Δscd6* strain in various culture conditions. Data plots represent mean ± SEM from of $n = 4$, where 'n' represents number of independent experiments. A two-tailed paired student $t$-test was used to calculate $P$-values. **C** Protein levels of Edc3-mCherry during conditions shown in A. Data plots represent mean ± SEM from of $n = 4$, where 'n' represents number of independent experiments. A two-tailed paired student $t$-test was used to calculate $P$-values. **D** Disassembly of endogenously tagged Edc3-mCherry granule in wild-type and *Δsbp1* after 1 h recovery. Scale Bar, 3 μm. Dotted line in the images represents cell boundary. **E** Quantification of endogenously tagged Edc3-mCherry granule per cell in -sodium azide, +sodium azide and recovery experiment presented in **D**. Data plots represent mean ± SEM from of $n = 3$, where 'n' represents number of independent experiments. A two-tailed paired student $t$-test was used to calculate $P$-values.

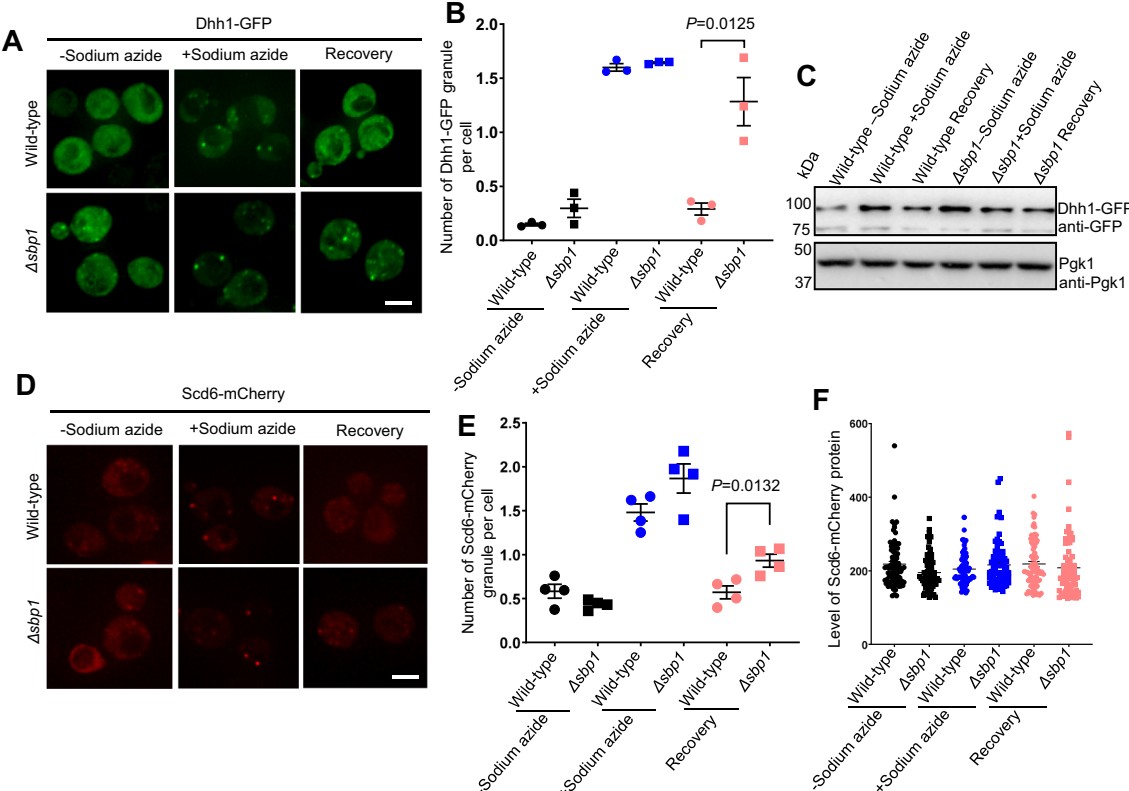

**Fig. 2 Disassembly of other P-body markers is defective in absence of Sbp1. A** Live-cell microscopy to observe disassembly of Dhh1-GFP as P-body marker. Scale Bar, 3 μm. **B** Graph depicting the number of Dhh1-GFP foci per cell in wild-type and *Δsbp1* strain. Data plots represent mean ± SEM from of *n* = 3, where '*n*' represents number of independent experiments. A two-tailed paired student *t*-test was used to calculate *P*-values. **C** Dhh1-GFP protein level during unstress, stress and recovery in wild-type and *Δsbp1* strain. **D** Live-cell microscopy to observe disassembly of Scd6-mCherry as P-body marker. Scale Bar, 3 μm. **E** Graph depicting the number of Scd6-mCherry foci per cell in wild-type and *Δsbp1* strain. Data plots represent mean ± SEM from of *n* = 4, where '*n*' represents number of independent experiments. A two-tailed paired student *t*-test was used to calculate *P*-values. **F** Protein level of Scd6-mCherry during unstress, stress, and recovery in wild-type and *Δsbp1* strain based on the measurement of mCherry signal using microscopy. Data plots represent mean ± SEM from of *n* = 3, where '*n*' represents number of independent experiments. A two-tailed paired student *t*-test was used to calculate *P*-values.

either Dhh1-GFP or Scd6-mCherry. The assembly of Dhh1-GFP granules was comparable to that of wild-type cells; however, the disassembly of Dhh1-GFP granules was significantly defective during recovery in *Δsbp1* strain as compared to wild-type (Fig. 2A, B). The disassembly defect was not due to increased protein levels of Dhh1-GFP (Fig. 2C). Similarly, the disassembly but not the assembly of Scd6-mCherry granules was affected in *Δsbp1* (Fig. 2D, E). The protein level of Scd6-mCherry remained unaffected in recovery as compared to stress conditions (Fig. 2F). We also checked the extent of disassembly after 2 h of recovery and observed that the disassembly of both Dhh1 and Scd6 granules were significantly defective even after 2 h of recovery (Supplementary Fig. 1A–D).

Since Scd6 has been reported to localize to both PBs and SGs[31], we tested if the Scd6 foci during recovery conditions in absence of Sbp1 were indeed PBs. Colocalization experiments using Edc3-mCherry*Δsbp1* strain revealed that a large majority of Scd6-GFP foci colocalized with Edc3-mCherry (Supplementary Fig. 2A, B). On the other hand, colocalization studies between Pab1-GFP (stress granule marker) and Scd6-mCherry in *Δsbp1* background expectedly did not show any colocalization since the Pab1 foci are not defective in disassembly in the absence of Sbp1 (Supplementary Fig. 2C, D). This indicates that the disassembly defective Scd6-mCherry foci constitute P-bodies. Based on the observations that disassembly of Edc3, Dhh1, and Scd6 granules is defective in absence of Sbp1, we conclude that PB disassembly is defective in *Δsbp1*.

**Edc3 granule disassembly defect in *Δsbp1* is not stress-specific**. The dynamics and composition of RNA granules differ among stress conditions[36]. Therefore, we tested whether P-body disassembly defect in *Δsbp1* strain was stress-specific. We analyzed disassembly of granules that are induced upon glucose deprivation in *Δsbp1*. Glucose deprivation leads to global translation repression and consequently RNA granule assembly[38]. As expected, the WT strain showed normal assembly and disassembly of stress granules (Pab1-GFP) and P-bodies (Edc3-mCherry) as reported earlier[6] (Fig. 3A; wild-type microscopy panel; Fig. 3B). Assembly of stress granules and P-bodies were comparable to that of wild-type in *Δsbp1* strain (Fig. 3A; *Δsbp1* minus glucose panel; Fig. 3B). Interestingly, the disassembly of P-bodies and not the stress granules was defective in *Δsbp1* (Fig. 3A; *Δsbp1* recovery panel; Fig. 3B). Protein levels of Edc3-mCherry during recovery were comparable to those under stress conditions (Supplementary Fig. 3A, B). This suggests that Sbp1 affects P-body disassembly under glucose deprivation conditions as well. Our results indicate that the role of Sbp1 in PB disassembly is general, thereby affecting multiple PB components under different stress conditions.

**RGG-motif of Sbp1 is necessary and sufficient for rescuing Edc3 granule disassembly defect in *Δsbp1*.** To confirm if the P-body disassembly defect observed in *Δsbp1* was indeed mediated by Sbp1, we complemented *Δsbp1* strain expressing endogenously tagged Edc3-mCherry with CEN plasmid (pRS315)

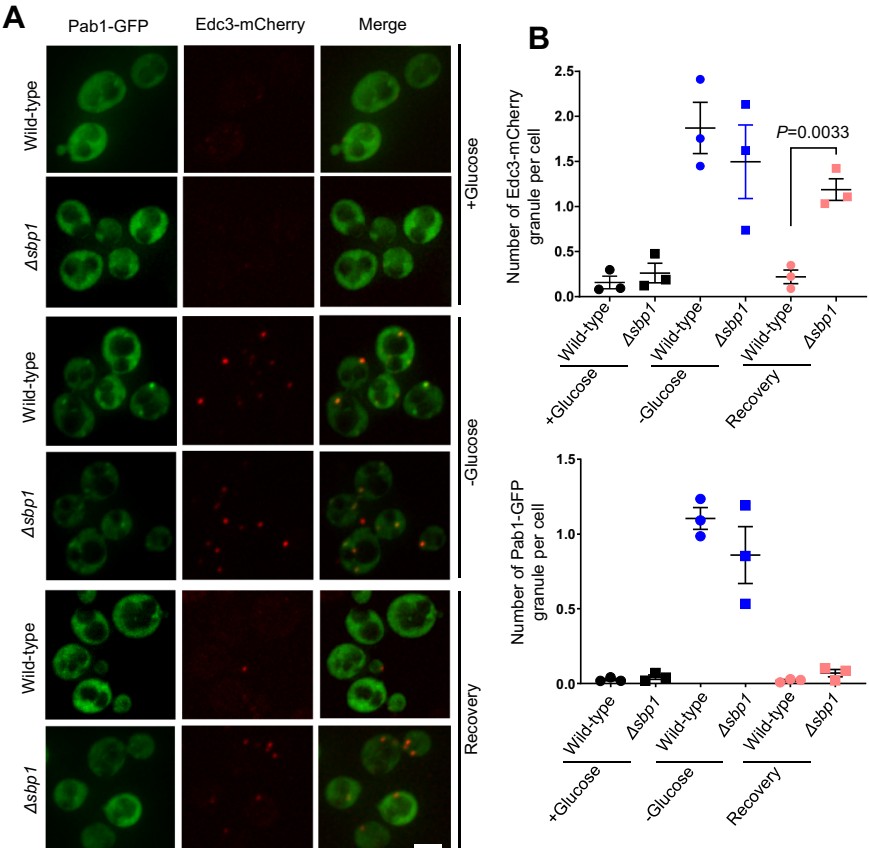

**Fig. 3 Edc3-mCherry granule disassembly defect is not stress-specific. A** Live-cell microscopy to observe disassembly of PBs and SGs formed upon stress condition in wild-type and *Δsbp1* strain. Scale Bar, 3 μm. **B** Graph depicting the number of PBs (top panel) and SGs (bottom panel) per cell in wild-type and *Δsbp1* strains in various culture conditions during granule recovery experiment. Data plots represent mean ± SEM from of *n* = 3, where '*n*' represents number of independent experiments. A two-tailed paired student *t*-test was used to calculate *P*-values.

expressing wild-type Sbp1 (pRS315-*SBP1*). We observed that the *Δsbp1* strain transformed with pRS315-*SBP1* showed disassembly of Edc3 granules formed during stress is comparable to the wild-type cells (Fig. 4A, B). We next investigated the role of RGG-motif in complementing the Edc3 granule disassembly defect. We created mutants of Sbp1 that were individually deleted for RGG motif, RRM1 domain, RRM2 domain, or both RRM1 + RRM2 domains in the pRS315-*SBP1* plasmid (Supplementary Fig. 4). The *SBP1ΔRGG* mutant was defective in complementing Edc3 granule disassembly (Fig. 4A, B; *Δsbp1*-pRS315-*SBP1ΔRGG* recovery panel) as compared to the full-length Sbp1 expressing in *Δsbp1* strain. Complementation using Sbp1 RGG-motif (lacking both RRM1 and RRM2 domains) indicated that the RGG-motif rescued the disassembly defect in a manner comparable to wild-type Sbp1 (Fig. 4A, B; *SBP1ΔRRM1ΔRRM2* microscopy panel). Mutants deleted for either RRM1 or RRM2 (Supplementary Fig. 5A) complemented the PB disassembly defect. These results suggest that the RGG-motif is necessary and sufficient for rescuing PB disassembly defects.

Recent studies have demonstrated the importance of arginine methylation in RNA granule assembly and disassembly[39–41]. Sbp1 has been reported to get arginine methylated by Hmt1[35]. This prompted us to test the ability of Arginine Methylation Defective (AMD) mutant of Sbp1 in complementing Edc3 disassembly. The AMD mutant of Sbp1 (wherein 13 arginine in the RGG-motif were converted to alanine; (Supplementary Fig. 4F) was used to demonstrate the role of AM in translation repression activity of Sbp1[35]. Complementation of the *Δsbp1* strain with the AMD mutant failed to rescue the disassembly

defect of P-bodies (Supplementary Fig. 5A). The AMD mutant is not defective in expression and folding[35] (Fig. 2D, H & J), ruling out the possibility that the phenotype is due to decreased protein levels. Thus, the AMD mutant is compromised in complementing Edc3 granule disassembly defect. Hmt1 (a PRMT1 homolog) is a predominant methyltransferase in yeast known to methylate Sbp1[35]. Interestingly, the disassembly of Edc3 granules in *Δhmt1* was comparable to the same in the wild-type strain (Supplementary Fig. 6A, B).

Deletion of Sbp1 leads to stress granule assembly defect (Fig. 1A) which is consistent with its role as a translation repressor. Sbp1 mutants such as with deletion of RGG-motif or the RRM1 domain and AMD mutant were observed to be defective in complementing stress granule assembly defect (Supplementary Fig. 5B), indicating a role of these domains in stress granule assembly.

**Sbp1 binds Edc3 in RGG-motif dependent manner.** Our results indicate a role of Sbp1 in PB disassembly, which is dependent on its RGG-motif. We hypothesized that a physical interaction between Sbp1 and Edc3 could facilitate disassembly of PBs; therefore, we tested the binding of Edc3 with Sbp1 and Sbp1ΔRGG in purified forms. We observed that Edc3 indeed binds to Sbp1 but not Sbp1ΔRGG (Fig. 5A). RNase A was added during protein purification as well as during the protein-protein interaction assay indicating that the interaction was independent of RNA. We conclude based on this observation that Sbp1 and Edc3 directly interact, and this interaction depends on the RGG-motif of Sbp1. Like Sbp1, Edc3 is a modular protein with Lsm,

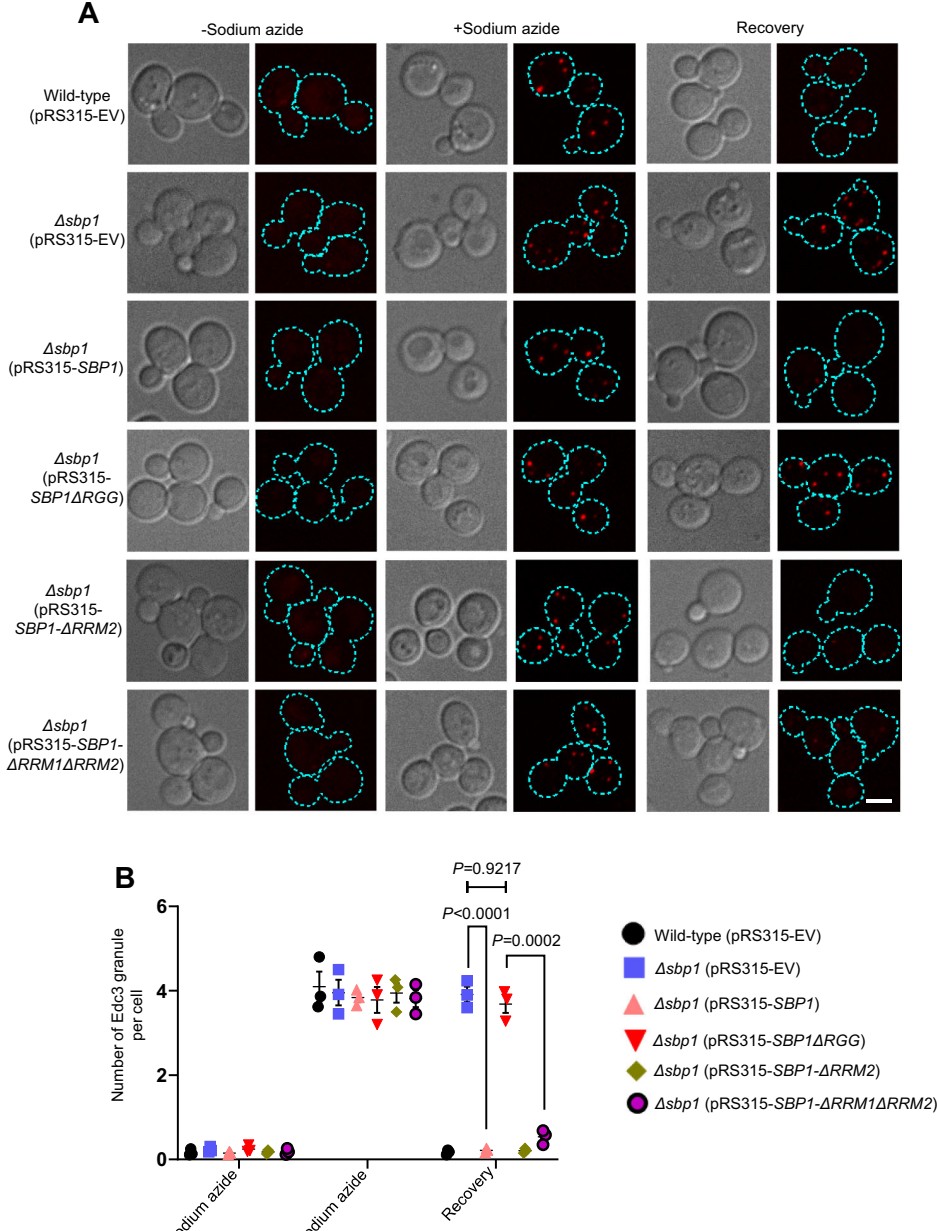

**Fig. 4 RGG motif is necessary and sufficient to rescue P-body disassembly defect. A** Live-cell microscopy to observe disassembly of PBs upon complementation with wild-type Sbp1 and mutants following sodium azide stress condition. Plasmid containing wild-type Sbp1-GFP and Sbp1 mutants were transformed in endogenously tagged Edc3-mCherry and Edc3m-Cherry*Δsbp1* cells as indicated. Cells were cultured till 0.35–0.4 $OD_{600}$ and incubated for 30 min with or without 0.5% (v/v) sodium azide at 30 °C. Subsequently, cells were pelleted and washed thrice with glucose-containing medium. For stress recovery, the resuspended cells were grown for an additional 1 h at 30 °C in media without sodium azide. Dotted lines in the images represent cell boundaries. Scale Bar, 3 µm. **B** Quantitation of the complementation experiment as described in **A**. Refer to Supplementary Fig. 4 for additional complementation results. Data plots represent mean ± SEM from of $n = 3$, where '$n$' represents number of independent experiments. A two-tailed paired student $t$-test was used to calculate $P$-values.

FDF, and Yjef-N domains. We, therefore, tested different domain deletion mutants of Edc3, which have previously been used for assessing interaction of Edc3 with different PB components[42], for interaction with recombinant Sbp1. Interaction studies revealed that purified Sbp1 could bind to both Lsm-FDF and Yjef-N domains but not the FDF domain alone (Fig. 5B, C). This interaction was independent of RNA as RNase A was added during protein purification as well as during the protein-protein interaction assay. The above results suggest that Sbp1 directly interacts with Edc3 via the Lsm and Yjef-N domains.

**Sbp1-Edc3 interaction competes with Edc3 self-association mediated by YjeF-N domain.** The C-terminal YjeF-N domain of Edc3 is central to its role as a P-body assembly factor. YjeF-N domain binds itself, leading to oligomerization of P-body mRNPs, thereby seeding higher-order structures[16]. Since Sbp1 binds Edc3 via its YjeF-N domain, we hypothesized that Sbp1 binding to Edc3 could affect Edc3 self-interaction. Interaction of full-length Edc3 with YjeF-N domain was tested in the presence of Sbp1. All the proteins used were in recombinant purified forms. Strikingly we observed that Edc3-YjeF-N interaction was compromised in

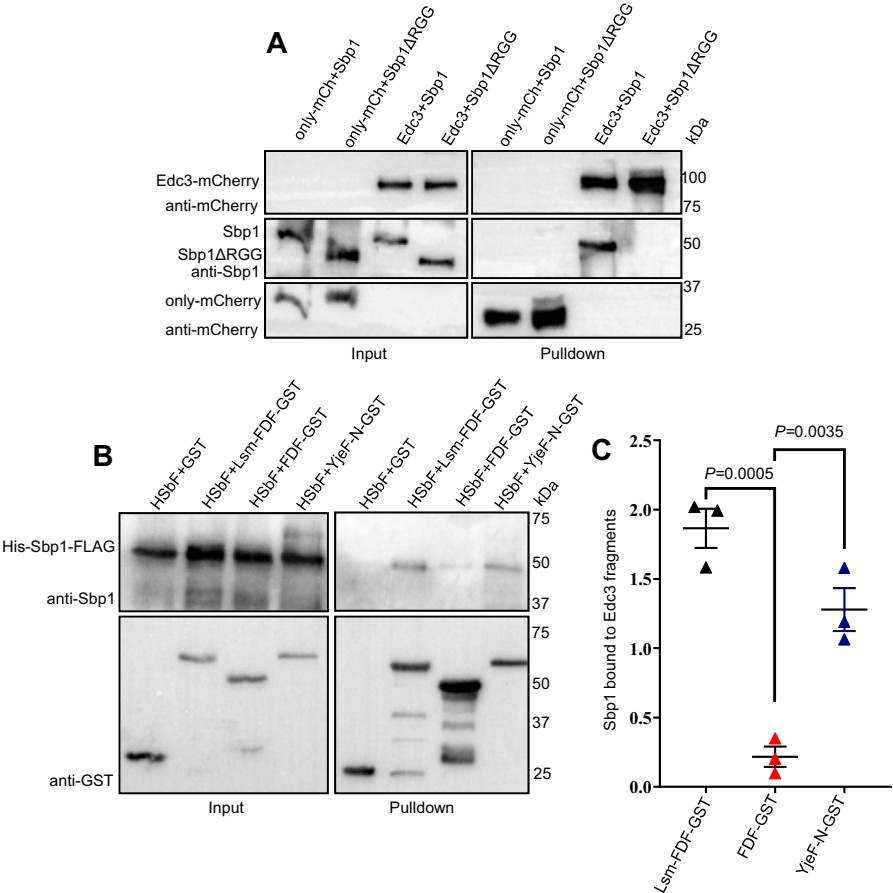

**Fig. 5 Sbp1 binds Edc3 in RGG-motif dependent manner. A** Full-length Edc3 binds to wild-type Sbp1 but not the ΔRGG mutant of Sbp1. Western blot depicting interaction of Edc3 with wild-type Sbp1 and Sbp1ΔRGG. 100 pmol of each of the purified proteins were taken for the binding assay. Data conclusions derived from of $n = 3$, where '$n$' represents number of independent experiments. **B** Western blot depicting interaction of Sbp1 with various fragments of Edc3. 200 pmol of each of the purified proteins were taken for the binding assay. **C** Quantitation of the results from three experiments as performed and presented in **B**. Amount of Sbp1 in pellet fractions was normalized by the GST-tagged protein in the same fraction. Data plots represent mean ± SEM from of $n = 3$, where '$n$' represents number of independent experiments. A two-tailed paired student $t$-test was used to calculate $P$-values.

the presence of Sbp1 (Fig. 6A, B). This result clearly indicated that Edc3-Sbp1 interaction could compete with Edc3 self-interaction.

**Purified Sbp1 leads to disassembly of Edc3-mCherry assemblies in vitro.** LLPS plays an important role in assembly of RNA granules[43]. Proteins with low complexity sequences, including Edc3, have been shown to undergo LLPS, which contributes to the assembly of RNA granules[44,45]. Although Edc3 from *S. pombe* has been recently studied to undergo LLPS, Edc3 from *S. cerevisiae* has not yet been reported to undergo LLPS. Therefore, we tested if purified Edc3 could phase separate to form higher-order assemblies. We observed that the purified Edc3-mCherry phase separated to form small assemblies (Fig. 7A). Strikingly the assemblies grew much larger in the presence of RNA and NADH (Fig. 7A, B).

Since the disassembly of Edc3 granules in cells was defective in the absence of Sbp1 and purified Edc3 self-interaction was compromised in the presence of Sbp1 (Fig. 6A, B), we hypothesized that Sbp1 could dissolve Edc3 assemblies by physically interacting with it. Upon addition of purified Sbp1, we indeed observed a reduction in the size of Edc3 assemblies. Importantly, decrease in the size of assemblies was dependent on the RGG-motif of Sbp1 as the addition of purified Sbp1ΔRGG did not affect Edc3 assemblies (Fig. 7C, D). We conclude, based on these results, that Edc3 can form higher-order assemblies which are augmented by RNA and NADH. Further, these assemblies are

dissolved upon incubation with purified Sbp1. This observation provides a mechanistic basis for the role of Sbp1 and its RGG-motif in P-body disassembly.

**EWSR1 aggregates persist in *Δsbp1* strain.** FET (FUS-Fused in Sarcoma; EWSR1-Ewings Sarcoma breakpoint region 1; TATA box binding protein Associated Factor 15) proteins have been implicated in neurodegenerative disorders such as Amyotrophic Lateral Sclerosis (ALS) and Fronto-Temporal Lobar Degeneration (FTLD). Mutations in EWSR1 have been identified in ALS patients[28]. EWSR1 has also been observed to co-accumulate with TAF15 and FUS in neuronal and glial cytoplasmic inclusions in FTLD patients[46]. Yeast has been successfully used as a model system to understand the role of FET protein toxicity[28,47,48]. Overexpression of EWSR1 in yeast leads to accumulation of EWSR1 protein in cytoplasmic aggregates[28]. Since Sbp1 was instrumental in PB disassembly, we decided to test the role of Sbp1 in modulating EWSR1 aggregates. EWSR1-YFP protein was expressed under galactose-inducible promoter in wild type and Sbp1 deletion strain which led to formation of EWSR1 aggregates (Fig. 8A). Strikingly upon shifting the cells to glucose media, EWSR1 aggregates are more persistent in absence of Sbp1 as compared to wild type (Fig. 8A, B). Consistent with this observation, overexpression of EWSR1 in absence of Sbp1 leads to a weak yet consistent growth defect (Fig. 8C). Even though EWSR1 protein levels decrease almost 2-fold in *Δsbp1* strain as compared

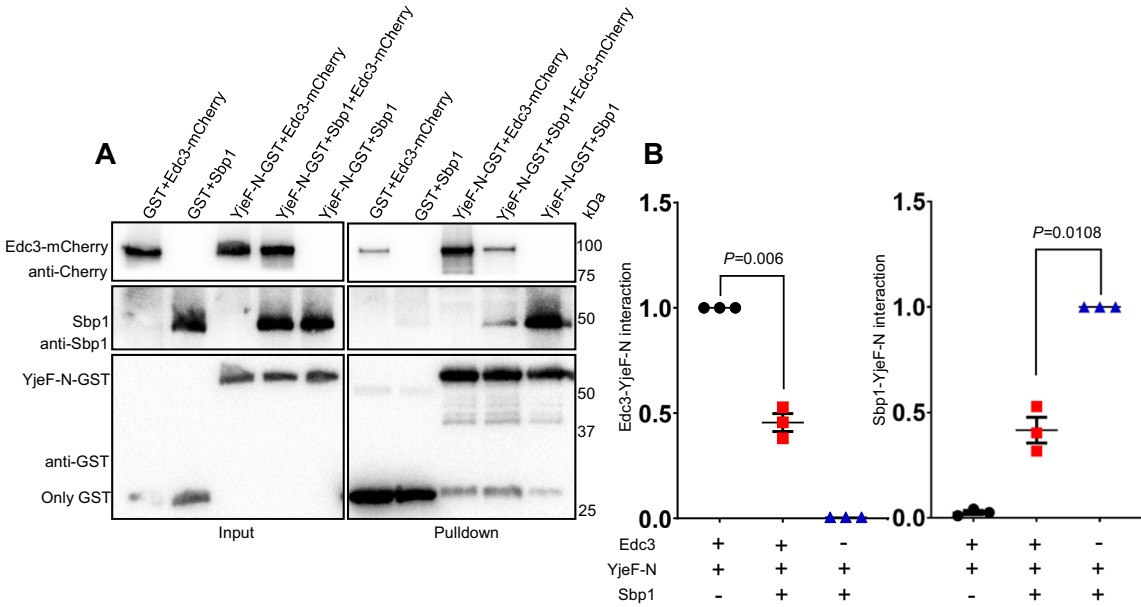

**Fig. 6 Presence of Sbp1 leads to decreased Edc3-YjeF-N interaction. A** Pull-downs followed by Western blot depicting competition between Sbp1-Edc3 interaction and Edc3 self-association. His-Sbp1-FLAG and Edc3-mCherry was purified using Ni-NTA chromatography. YjeF-N-GST fragment and only GST was purified with glutathione sepharose beads. Interaction studies using 100 pmol of each protein was carried out in 1X protein–protein interaction buffer (PPIB). See Materials and Methods section for details. **B** Quantitation of Western blot data presented in **A**. Left panel represents graph measuring Edc3-YjeF-N interactions whereas the panel on right represents the Sbp1-YjeF-N interactions. Data plots represent mean ± SEM from of $n = 3$, where '$n$' represents number of independent experiments. A two-tailed paired student $t$-test was used to calculate $P$-values.

to wild type (Fig. 8D, E), the presence of this phenotype suggests that comparable EWSR1 levels in $\Delta sbp1$ strain as the wild type would lead to a stronger growth defect. Taken together, these observations suggest that Sbp1 contributes to the clearance of heterologous aggregates of disease-relevant proteins as well.

## Discussion

This study makes two important conclusions. It identifies a P-body disassembly factor in the form of Sbp1 and implicates a low complexity sequence (RGG-motif) in promoting disassembly. The following observations support these conclusions: (1) Disassembly of P-body is defective upon recovery from stress based on the results using three PB markers, (2) Complementation of $\Delta sbp1$ strain by wild-type $SBP1$ but not $SBP1\Delta RGG$ rescues the P-body disassembly defect, (3) RGG-motif is sufficient to complement the disassembly defect, (4) Sbp1 directly interacts with Edc3 in RGG-motif dependent manner, (5) Sbp1-Edc3 interaction competes with Edc3 self-interaction, (6) Purified Sbp1 but not Sbp1ΔRGG dissolves Edc3 assemblies in vitro, (7) EWSR1 aggregates persist longer in the absence of Sbp1 and (8) Overexpression of EWSR1 in absence of Sbp1 leads to a growth defect.

Our data proposes a simple yet lucid model to explain the mechanistic basis of the role of Sbp1 in disassembly. Sbp1 directly binds Edc3 (Fig. 5). The RGG-motif of Sbp1 and YjeF-N domain of Edc3 participate in this interaction (Fig. 5). The competition between Edc3-Sbp1 and Edc3 self-interaction underlines the mechanism enabling Sbp1 to disassemble P-bodies (Fig. 6). This study identifies Sbp1 as a new direct binding partner of Edc3, which promotes P-body disassembly via competing with Edc3 self-interaction.

An obvious yet interesting question is the temporal regulation of this interaction. It may be hypothesized that the recruitment of Sbp1 to Edc3 assemblies may be promoted during recovery by a bridging adapter protein. It is also possible that Sbp1 and/or Edc3 undergo post-translational modification(s) during recovery conditions that enable their interaction with each other. Importantly,

the AMD (Arginine Methylation Defective) mutant of Sbp1 fails to complement the Edc3 disassembly defect (Supplementary Fig. 5A), indicating that arginine methylation (AM) could be important for its role in disassembly. It is noteworthy that the deletion of Hmt1 (arginine methyltransferase) does not perturb the ability of Sbp1 to promote Edc3 disassembly (Supplementary Fig. 6). This could be due to backup methyltransferase activity in the absence of Hmt1 as observed earlier for Sbp1, Scd6, and other proteins[32,35,49]. The precise role of AM in PB disassembly by Sbp1 will be an important future direction.

It is intriguing that translation repressor protein Sbp1 itself localizes to P-bodies and promotes its disassembly. It is currently unclear if the role of Sbp1 in repression and granule disassembly is related. It must be noted that both the functions of Sbp1 depend on its RGG-motif, indicating that the two could be related. Absence of a related RGG-motif containing protein Scd6 does not lead to disassembly defect (Fig. 1A, B), indicating that RGG-motifs of Scd6 and Sbp1 are functionally different, although both interact with eIF4G to promote translation repression. It would be interesting to understand the basis of this functional difference between RGG-motifs in different RNA-binding proteins. Interestingly Scd6 localizes to both PBs and SGs. We believe that the relatively weaker disassembly defect observed for Scd6 foci (Fig. 2D, E) represents Scd6 localizing to SGs, as disassembly of SG is not defective in absence of Sbp1. This is confirmed by our observation that the disassembly defective Scd6 foci localize with P-bodies and not stress granules (Supplementary Fig. 2). The specific role played by the Lsm domain of Edc3 in binding to Sbp1 remains to be addressed in future.

Assembly of RNA granules during stress can be defective in the absence of translation repressors[31]. Sbp1 localizes to RNA granules, and its deletion decreases assembly of stress granules (Fig. 1A, B). Interestingly mutants lacking RGG-motif or the RRM2 domain are defective in complementing the stress granule assembly defect (Supplementary Fig. 5B). Since stress granules arise from pre-formed PBs in yeast[6], it is likely that PB

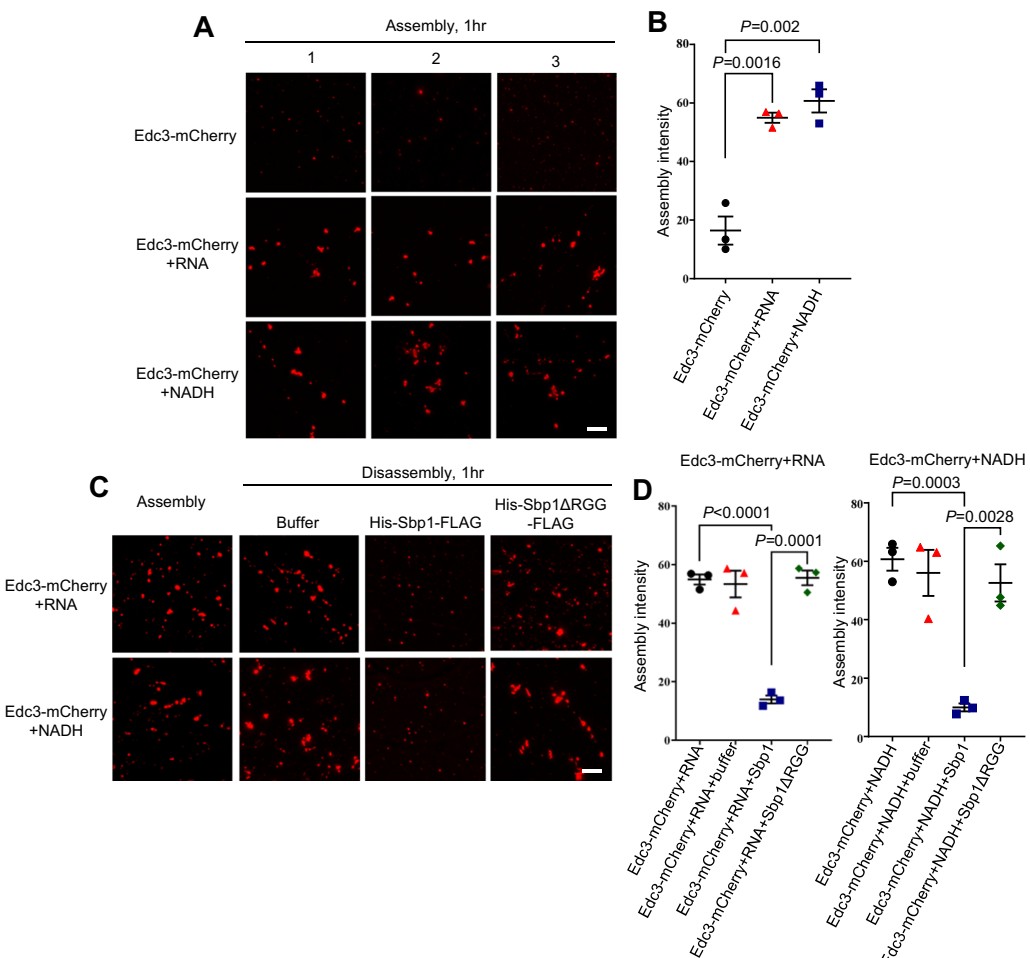

**Fig. 7 Purified Sbp1 disrupts Edc3 assemblies in RGG-motif dependent manner. A** Purified Edc3-mCherry (10 μM) were kept at 30 °C for 1 hr to phase-separate in LLPS buffer (150 mM KCL; 30 mM HEPES-KOH pH 7.4; 2 mM MgCl$_2$) in presence of RNA or NADH as indicated. 1, 2, and 3 indicate representative images from three independent experiments. Scale Bar, 10 μm. **B** Quantitation for A. Data plots represent mean ± SEM from of $n = 3$, where '$n$' represents number of independent experiments. A two-tailed paired student $t$-test was used to calculate $P$-values. **C** After 1 h, the phase-separated assemblies were subjected to 10 μM of Sbp1 or Sbp1ΔRGG protein and incubated again at 30 °C for another 1 h followed by microscopy at each time point during assembly and disassembly. The images were taken using DeltaVision DV Elite microscope. Scale Bar, 10 μm. **D** Quantitation of data presented in **C**. Data plots represent mean ± SEM from of $n = 3$, where '$n$' represents number of independent experiments. A two-tailed paired student $t$-test was used to calculate $P$-values.

disassembly defect in Δ*sbp1* could affect the exchange of mRNPs, contributing to the stress granule assembly defect. Understanding the connection between PB disassembly defect and stress granule assembly defect in absence of Sbp1 will be a future research direction.

Interestingly, we observe that NADH and RNA can independently augment formation of Edc3-mCherry assemblies (Fig. 7). This is consistent with the report that *S. pombe* Edc3 phase separation is promoted by RNA[44]. The FDF and the YjeF-N domains of Edc3 interact with RNA, which facilitates phase separation. The role of NADH in Edc3 assembly has not been reported. It is known that human Edc3 binds NADH, and both human and yeast Edc3 can chemically modify NAD[50]. Mutants of yeast Edc3 predicted to be defective in binding NADH were defective in localizing to P-bodies[50]. Our result provides a strong basis for focusing on the role of NADH in Edc3 function in vivo.

Previous reports suggest that autophagy plays an important role in stress granule clearance. VCP, a human ortholog of yeast Cdc48, is required for efficient stress granule clearance through granulophagy. Downregulation or chemical inhibition of VCP in HeLa cells led to a significant defect in stress granule clearance in mammalian cell culture[51]. ZFAND1 is a human ortholog of yeast Cuz1, a protein implicated in arsenite response in yeast[52,53]. ZFAND1 was reported to trigger proteasomal degradation of stress granules during arsenite-induced stress and its clearance during recovery. siRNA depletion of ZFAND1 resulted in granule clearance defect 2 h after recovery from arsenite stress[54]. Cuz1 deletion leads to defective SG clearance upon sodium arsenite treatment. Therefore, we wondered if Cuz1 could affect P-body disassembly upon sodium azide stress. We observed that the assembly of stress granule and P-bodies were defective in Δ*cuz1* as compared to wild-type (Supplementary Fig. 7). However, Δ*cuz1* did not affect PB and SG disassembly (Supplementary Fig. 7). This suggests that Cuz1 does not play a role in clearance of SGs and PBs that arise in response to sodium azide stress.

Mutations in proteins with low complexity sequences such as FUS, EWSR1, and TDP43 are implicated in neurodegenerative disorders like ALS[26,55–57]. These proteins form higher-order assemblies which can be toxic[47]. Persistence of EWSR1 aggregates in absence of Sbp1 (Fig. 8A, B) is consistent with our result demonstrating defective granule disassembly in absence of Sbp1.

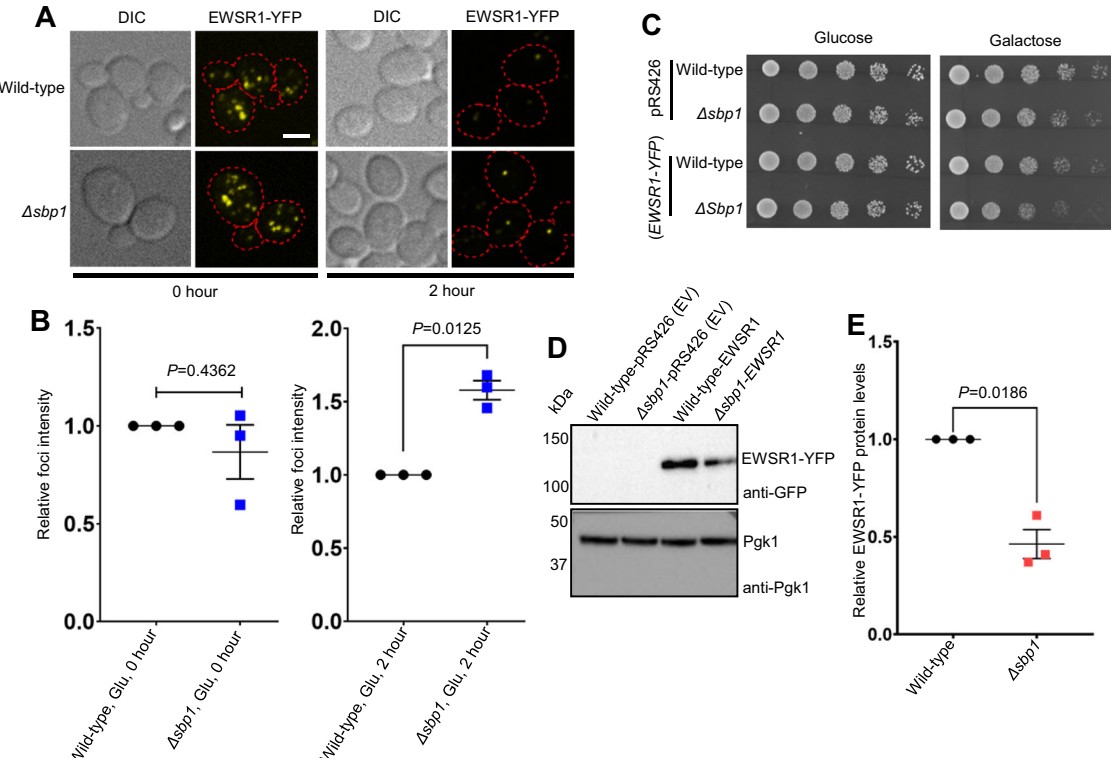

**Fig. 8 EWSR1 granules persist in Δ*sbp1*. A** Wild-type and Δ*sbp1* strain transformed with plasmid expressing galactose-inducible EWSR1-YFP (pAG426) plasmid were inoculated at $OD_{600}$ 0.15 and cultured at 30 °C till $OD_{600}$ 0.4–0.6. Cell were washed 3X with SD media supplemented with 2% galactose and resuspended finally in media with 2% galactose for 1.5 h to induce the expression of EWSR1-YFP. Cells were then washed 3X with SD media supplemented with 2% glucose and cultured for 2 h in media with 2% glucose to monitor EWSR1-YFP foci. Images were taken at 0 h and 2 h post-induction. Scale Bar, 3 μm. **B** Quantitation for EWSR1 foci at 0- and 2-h post-induction as presented in A. EWSR1-YFP foci intensities were quantified using the Measure tool in Fiji ImageJ version 1.53f51. EWSR1-YFP foci intensities in the two conditions (0 h and 2 h) were normalized with the wild type foci intensities in the respective conditions and plotted using GraphPad Prism 4 to quantify the relative foci intensities in Δ*sbp1* strain. Data plots represent mean ± SEM from of *n* = 3, where '*n*' represents number of independent experiments. A two-tailed paired student *t*-test was used to calculate *P*-values. **C** Growth assay for wild type and Δ*sbp1* transformed with empty vector and plasmid expressing galactose-inducible EWSR1-YFP. Cells were serially diluted (starting from $OD_{600}$ = 1.0) and spotted on agar plates containing SD media without uracil supplemented with either 2% glucose or 2% galactose. The plates were incubated at 30 °C and images were acquired after 24 and 36 h for glucose and galactose plates respectively. **D** Western blot analysis to assess EWSR1-YFP protein levels for the overexpression growth assays in wild-type and Δ*sbp1* presented in (**C**). **E** Quantitation of the western blot analysis presented in (**D**). Data plots represent mean ± SEM from of *n* = 3, where '*n*' represents number of independent experiments. A two-tailed paired student *t*-test was used to calculate *P*-values.

This result extends the role of Sbp1 in granule disassembly towards heterologous proteins, indicating that the role of RGG-motif in disassembly could be a general phenomenon. This idea opens exciting avenues of research. Future experiments will identify the mechanism underlying clearance of EWSR1 aggregates mediated by RGG-motif containing low complexity sequences. It must be noted that the evidence in literature suggests a role for Sbp1 in modulating FUS toxicity. Sbp1 was identified as a multi-copy suppressor of FUS-related toxicity in yeast cells[47,48]. Our observation that EWSR1 overexpression leads to a growth defect in the absence of Sbp1 (Fig. 8C) is consistent with the above reports. It provides indication of the consequence of disassembly defects of disease-relevant protein aggregates in absence of Sbp1.

Overall, our report identifies a bonafide PB disassembly factor that functions through low-complexity sequences. This report paves the way to screen for and identify other disassembly factors of PBs and RNA granules in general. Such factors will enhance our understanding of the role of RNA granules in the regulation of mRNA fate under normal conditions and diseased states. Contrary to the general understanding, our report highlights a new role of LC sequences in granule disassembly.

## Methods

**List of strains used in this study**. See Supplementary Table 1 for strain descriptions.

**List of plasmids used in this study**. See Supplementary Table 2 for plasmid descriptions.

**List of antibodies used in this study**. See Supplementary Table 3 for antibody descriptions.

**List of primers used in this study**. See Supplementary Table 4 for primer descriptions.

**Yeast cell culture for microscopy experiments**. Overnight culture of yeast strains (transformed with the indicated plasmid) grown in appropriate synthetically defined (SD) drop-out growth medium was diluted to $OD_{600}$ 0.1 and grown at 30 °C until they reached an $OD_{600}$ of 0.4–0.5. Cultures were then incubated for 30 min with 0.5% sodium azide v/v (stock concentration of sodium azide was 20% w/v) at 30 °C or with an equal volume of water. Subsequently, the cells were pelleted by centrifugation (3234 g, 10 s, RT) and washed thrice with glucose-containing SD medium. For stress recovery, the resuspended cells were grown for an additional 1 h (or 2 h for experiments in Supplementary Fig. 1) at 30 °C in media without sodium azide. 1.5 mL cells were collected by centrifugation (14,000 g, 12 s, RT), supernatant was removed, and the cells were resuspended in 20 μl of residual media. Cells were spotted on coverslips for microscopy at room

temperature. All processing steps, including microscopy after growth of cells were carried out at a controlled room temperature of 23–25 °C.

For glucose starvation experiment, overnight cultures of yeast strains (transformed with the indicated plasmid) in SD Ura- yeast growth medium were diluted to $OD_{600}$ 0.1 and grown at 30 °C until they reached an $OD_{600}$ of 0.4–0.5. Cultures were divided into two equal parts and washed, one part with glucose-containing media and the other part with glucose-deprived media. Cultures were then incubated for 30 min in the presence and absence of 2% glucose at 30 °C. Subsequently, the cells were pelleted by centrifugation (3234 g, 10 s, RT) and washed twice with SD ura- medium with glucose. For stress recovery, the resuspended cells were grown for an additional 1 h at 30 °C. Microscopy was carried out in each of the conditions mentioned above at room temperature (23–25 °C). For microscopy, 1.5 mL cells were collected by centrifugation (14,000 g, 12 s, RT). Supernatant was removed, and the cells were resuspended, then immobilized on coverslips for microscopy at room temperature.

Disassembly experiment for EWSR1 was carried out by inoculating the wild-type and Δsbp1 strains transformed with plasmid expressing galactose-inducible EWSR1-YFP (pAG426) at $OD_{600}$ 0.15 and cultured at 30 °C till O.D.$_{600}$ 0.4–0.6. Cells were washed thrice with SD media supplemented with 2% galactose and resuspended finally in media with 2% galactose for 1.5 h to induce the expression of EWSR1-YFP. Cells were then washed thrice with SD media supplemented with 2% glucose and cultured for 2 h in media with 2% glucose to monitor disassembly of EWSR1-YFP foci. Images were taken at 0 h and 2 h post-induction stop.

**Microscopy**. All the microscopy experiments, including imaging steps, were carried out at a controlled room temperature (23–25 °C). All images were acquired using a Deltavision RT microscope system running softWoRx 6.1.3 software (Applied Precision, LLC), using an Olympus 100X, oil-immersion 1.516 NA objective. Exposure time and transmittance settings for Green Fluorescent Protein (GFP) channel were 0.2 s and 32%, and for mCherry channel were 0.3 s and 32% respectively (for mCherry expressed on plasmid). Exposure time and transmittance settings for endogenously expressed mCherry (Fig. 1D, E) were 0.8 s and 100% respectively. This was required as the signal intensity of Edc3 endogenously tagged with mCherry is weaker than Edc3-mCherry expressed on plasmid. Images were collected as 512 × 512-pixel files with a CoolSnapHQ camera (Photometrics) using 2 × 2 binning for yeast. All microscopy images were deconvolved using softWoRx 6.1.3, the software used by DeltaVision[58]. Granule counting was performed using Fiji ImageJ Version 1.53f51. For each experiment, >100 cells were considered for granule counting manually. Data from three independent experiments were used for quantitation, and statistical significance was calculated using non-parametric student t-test.

Quantitation for EWSR1-YFP foci intensities was performed using the 'Measure tool' in Fiji ImageJ version 1.53f51. Intensities of all the foci were measured. Exposure time and transmittance settings for YFP channel were 0.2 s and 32% respectively. Images were collected as 512 × 512-pixel files with a CoolSnapHQ camera (Photometrics) using 2 × 2 binning for yeast. All microscopy images were deconvolved using softWoRx 6.1.3, the software used by DeltaVision[58]. Data plots represented are mean ± SEM from of n = 3, where 'n' represents number of independent experiments. A two-tailed paired student t-test was used to calculate P-values. EWSR1-YFP foci intensities in the two conditions (0 h and 2 h) were normalized with the wild type foci intensities in the respective conditions and plotted using GraphPad Prism 4.

**Complementation experiment**. Complementation experiments presented in Fig. 4 were performed using endogenously tagged Edc3-mCherry-WT, and Edc3-mCherry Δsbp1 strains transformed with CEN plasmid (pRS315) expressing wild-type Sbp1-GFP or its mutants (Sbp1ΔRGG-GFP, Sbp1ΔRRM2-GFP, and Sbp1 ΔRRM1ΔRRM2-GFP)[59].

Complementation experiments presented in Supplementary Fig. 5 were performed using wild-type and Δsbp1 strain transformed with plasmid expressing Pab1-GFP as stress granule marker and Edc3-mCherry as P-body marker and CEN plasmid (pRS315) expressing either wild-type SBP1 or its mutants (Sbp1ΔRGG, Sbp1ΔRRM1, Sbp1ΔRRM2, and Sbp1-AMP). To clone wild-type SBP1 gene, SBP1 ORF was amplified using specific primers from S. cerevisiae genomic DNA along with its promoter. The plasmid was digested with SmaI resulting in blunt end plasmid DNA. The amplified SBP1 was then ligated to the digested pRS315 plasmid using T4 DNA ligase (Thermo). The mutants were created by the same method for site-directed mutagenesis using specific primers designed using Agilent Quick-change Primer design web tool.

**Growth assay**. Growth assay was performed for wild type and Δsbp1 transformed with empty vector and plasmid expressing galactose-inducible EWSR1-YFP. Cells were serially diluted (starting from $OD_{600}$ 1) and spotted on agar plates containing SD media without uracil supplemented with either 2% glucose or 2% galactose. The plates were incubated at 30 °C, and images were acquired after 24 and 36 h for glucose and galactose plates, respectively.

**Western analysis**. For performing western analysis to estimate Edc3-mCherry, Dhh1-GFP levels in wild-type and Δsbp1 under unstressed, stressed, and recovery

conditions, an aliquot of cells were collected by centrifuging at 3234 g for 1 min at room temperature for each condition while doing microscopy experiments. Cells were then lysed in 100 µl lysis buffer containing 50 mM Tris–Cl pH 7.5, 50 mM NaCl, 2 mM MgCl₂, 0.1% Triton-X100, 1 mM β-Mercaptoethanol, 1× cComplete mini-EDTA-free Protease Inhibitor Cocktail (Roche, catalog no. 04693132001) and lysed by vortexing at 4 °C in bead-beater with glass beads. Unbroken cells and debris were removed by centrifugation at 2800 g for 5 min at 4 °C, followed by a 1-min spin at 15,800 g to remove any protein aggregates. 100 µg of total protein was loaded onto SDS-PAGE gels. Western analysis was performed using anti-mCherry antibody (Abcam, cat. ab167453), anti-GFP (Santa Cruz, catalog no. sc-9996; 1:1000 dilution), and anti-PGK1 (Abcam, catalog no. ab113687; dilution 1:1000). Western data analysis was done using Biorad ImageLab version 6.0.1.

**Protein expression and purification**. For in vitro pull-down experiments, proteins were purified from E. coli BL21 by batch purification using either glutathione sepharose (GE Healthcare, Chicago, IL, USA, catalog no. 17075604) or Ni-NTA agarose (ThermoFisher Scientific, catalog no. 88222). Cell pellet was resuspended in lysis buffer along with addition of lysozyme (10 µg mL⁻¹), DTT (1 mM), PMSF (2 mM) cOmplete mini-EDTA-free Protease Inhibitor Cocktail (Roche, Basel, Switzerland), and RNase A (1 mg mL⁻¹) for 20 min. Lysis was performed using sonication, and debris was separated by centrifuging at 21,000 g for 15 min at 4 °C. Lysate was allowed to bind to beads for 1 h on nutator at 4 °C. Washes were performed on nutator for 10 min at 4 °C with Ni-NTA wash buffer (300 mM NaCl, 150 mM NaH₂PO₄, 20–40 mM imidazole) for His-tagged purification and with 1X PBS for GST-tagged purification. For His-tagged proteins, elution was done with 250 mM of imidazole (SRL cat# 61510), while for GST-tagged purifications, proteins were kept bead-bound in storage buffer (10 mM Tris Base pH 7.0, 25 mM NaCl, 1 mM DTT, 20% Glycerol) at 4 °C for immediate use in Fig. 5B. For pull-down in Fig. 6A GST-tagged proteins were purified using 50 mM Tris-HCl, 10 mM reduced glutathione, pH 8.0. Purified His-tagged protein was concentrated and dialyzed into 10 mM Tris–Cl pH 7.5, 100 mM NaCl, 20% glycerol, and 1 mM DTT in the cold room. Western analysis was performed using anti-GST (CST, catalog no. 2624; 1:1000 dilution), anti-Sbp1 antibody[35] (1:10000 dilution), and anti-FLAG antibody (Sigma, catalog no. F3165, 1:2000 dilution) for checking the purity of protein and then used for binding studies.

For Edc3 assembly assay, His-Edc3-mCherry. His-Sbp1-FLAG and His-Sbp1ΔRGG-FLAG proteins were purified from E. coli BL21 by batch purification using Ni-NTA agarose (ThermoFisher Scientific, catalog no. 88222). Cell pellet was resuspended in L-arginine lysis buffer (50 mM Tris–Cl pH 7.4; 500 mM NaCl; 10% Glycerol; 5 mM β-mercaptoethanol; 200 mM L-arginine pH 7.4) along with the addition of lysozyme (10 µg mL⁻¹), DTT (1 mM), PMSF (2 mM) and cOmplete mini-EDTA-free Protease Inhibitor Cocktail (Roche, Basel, Switzerland), RNase A (1 mg mL⁻¹) for 20 min. Lysis was performed using sonication (SONICS, VibraCell; Pulse rate of 10 s ON and 10 s OFF for 90 s)[32]. Debris was separated by centrifuging at 21,000 g for 15 min at 4 °C. Lysate was allowed to bind to beads overnight on nutator at 4 °C. Washes were performed on nutator for 10 min at 4 °C with Ni-NTA wash buffer (300 mM NaCl, 150 mM NaH₂PO₄, 20–40 mM imidazole). Elution was performed with 250 mM of imidazole (SRL, cat. 61510). Purified His-tagged protein was concentrated and dialyzed into 10 mM Tris–Cl pH 7.5, 100 mM NaCl, 20% glycerol, and 1 mM DTT in the cold room. Coomassie brilliant blue staining was performed to check the purity of the purified protein. Western blot analysis was performed with anti-mCherry antibody (Abcam, cat. ab167453) and anti-Sbp1 antibody.

**In vitro pull-down**. For recombinant protein pull-downs, 100 pmol (unless indicated otherwise) of purified proteins were used. Purified protein was incubated with immobilized GST-tagged protein to glutathione sepharose beads (GE Healthcare) in Fig. 5B or purified proteins were incubated with glutathione sepharose beads (GE Healthcare) in Fig. 6A at 4 °C for binding reactions (2 h). The binding buffer for glutathione pull-downs contained 50 mM HEPES pH 7, 100 mM NaCl, 1 mM DTT, 2 mM MnCl₂, 2 mM MgCl₂, 1% Triton-X100, 10% glycerol, 0.25 mg mL⁻¹ RNase A and 10 mg mL⁻¹ BSA. The beads were washed thrice with binding buffer, and 10 µL of SDS-PAGE loading dye was added to the beads and analyzed by SDS-PAGE followed by western blotting.

For RFP (Edc3-mCherry) pull-downs, RFP magnetic agarose beads (Chromotek, Rtma-20) were initially blocked with 5% skimmed milk in wash buffer (50 mM HEPES pH 7, 100 mM NaCl, 1 mM DTT, 2 mM MnCl₂, 2 mM MgCl₂, 1% Triton-X100, 10% glycerol) for 1 h. After blocking, the magnetic agarose beads were washed thrice using wash buffer on magnetic stand (Magna GrIP™ Rack, Cat. 20-400). 100 pmol of respective purified proteins were incubated with magnetic RFP agarose beads at 4 °C for binding reactions (for 2 h). The binding buffer for RFP pull-downs contained 50 mM HEPES pH 7, 100 mM NaCl, 1 mM DTT, 2 mM MnCl₂, 2 mM MgCl₂, 1% Triton-X100, 10% glycerol, 0.25 mg mL⁻¹ RNase A. The beads were washed thrice with wash buffer followed by the addition of 10 µL of SDS-PAGE loading dye to beads and analyzed by SDS-PAGE followed by western blotting.

**Edc3 assembly assay**. Reactions were performed in 1.5 mL micro-centrifuge tubes. Proteins were diluted to 10 µM in assembly buffer containing 150 mM KCl,

30 mM HEPES-KOH pH 7.4, 2 mM MgCl₂, 1 µg of total RNA or 0.2 mM NADH in a reaction of 30 µL and kept at 30 °C untouched for liquid-droplet formation for 1 h. The reactions were centrifuged at 200 g for 2 min, and the supernatant was discarded very gently, leaving 10 µL of reaction followed by microscopy.

For analyzing the impact of Sbp1 on Edc3 assemblies, purified Sbp1, Sbp1ΔRGG, BSA, or buffer were added at 10 µM concentration in the above pre-formed droplet reactions after 1 h of assembly and incubated for another 1 h at 30 °C. The reactions were centrifuged at 200 g for 2 min, and supernatant was discarded very gently, leaving 10 µL of reaction followed by microscopy at room temperature (23–25 °C). Microscopy was performed using a DeltaVision RT microscope system running softWoRx 6.1.3 software (Applied Precision, LLC), using an Olympus 100X, oil-immersion 1.516 NA objective. Exposure time and transmittance settings for Green Fluorescent Protein (GFP) channel were 0.2 s and 32%, and for mCherry channel were 0.3 s and 32% respectively. Images were collected as 512 × 512-pixel files with a CoolSnapHQ camera (Photometrics) using 2 × 2 binning. All images were deconvolved using standard softWoRx 6.1.3 deconvolution algorithms for granule counting and data analysis. Intensity of the Edc3-mCherry assemblies was quantified using Fiji ImageJ software version 1.53f51.

**Reporting summary**. Further information on research design is available in the Nature Research Reporting Summary linked to this article.

## Data availability

All data that support the findings of this study are available within this article, Supplementary Information, or available from the corresponding author upon reasonable request. Source data underlying Figs. 1–8 and Supplementary Figs. 1–7, are provided as a Source data file with this paper.

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

## Acknowledgements

We are indebted to Roy Parker for providing us various plasmids. We thank Karsten Weiss and Aaron Gitler for their generous gift of Dhh1-GFP and EWSR1-YFP plasmids respectively. We thank Jeff Wilusz, Ross Buchan, and Angela Hilliker for reading the manuscript and providing insightful comments. We thank Rajyaguru lab members for their constant inputs, support, and encouragement. PIR thanks DBT/Wellcome Trust India Alliance Fellowship/Grant [IA/I/12/2/500625] and DBT India grant [BT/PR40106/BRB/10/1918/2020] for supporting this research work. We thank the DBT-IISc partnership program and DST-FIST program for infrastructure support. RR and NB thank DBT and IISc for their respective fellowships. IAK and GD thank DBT for project assistantship.

## Author contributions

Conceptualization and hypothesis—PIR and RR; Experimental design—PIR and RR; Experimentation—RR, GD, and IAK; Data interpretation—PIR, RR, GD, and IAK; Manuscript writing (first draft)—PIR, Subsequent draft review and editing—PIR, RR, and GD; NB created the pRS315 (*SBP1ΔRRM1*) and pRS315(*SBP1ΔRRM2*) mutant constructs from pRS315(*SBP1*) created by RR.

## Competing interests

The authors declare no competing interests.
