## [Peer Review File · Nature Communications]

Low complexity RGG-motif sequence is required for
Processing body (P-body) disassemblyREVIEWER COMMENTS

Reviewer #1 (Remarks to the Author):

Roy et al. 2021 aims to expand the role of low complexity sequences in the dynamic liquid-liquid phase separation (LLPS) of processing body (PB) granules. The low complexity regions that make up intrinsically disordered regions (IDRs) have been previously implicated in LLPS, and this work could potentially expand on the role of how PB formation and dissolution functions as a dynamic regulator of cellular processes. Investigators attempt to characterize an interesting phenotype in which Sbp1 null mutants form granules under stress conditions that fail to disassemble once placed in recovery conditions. However, enthusiasm for this work is greatly diminished by a general lack of evidence to support the author conclusions. Moreover, there are some significant issues in the experimental design. Based on this, the conclusions of the manuscript are premature.

Major issues

The authors use transgenic plasmids with reporter genes that label PB and stress granule (SG) components in two different mutant strains for granule-associated proteins that both contain a low complexity RGG repetitive sequence, Sbp1 and Scd6. They propose the novelty of a low complexity sequence in the disassembly of P bodies seen in their Sbp1 mutant; however they the authors do not accurately distinguish between polar bodies and stress granules. As seen in any figure containing a merge of fluorescently labeled Pab1 and Edc3, there are numerous granules that overlap. Colocalization was not assessed or addressed at any point in the manuscript. This is also made apparent upon the recognition that one gene of interest, Scd6, is present in both PBs and SGs. In figure 2D, an Scd6-mCherry reporter transgene is used to assess the disassembly phenotype in Sbp1 null mutants. There is no discernment between which of the Scd6-labeled granules is a SG or PB, contradicting the conclusions from the first figure in which Sbp1 mutants only show a disassembly for PBs. Sbp1 deletion mutants, as well as some of the other partial deletion constructs also show a defect in SG assembly, however there is no attempt to address or characterize Sbp1 involvement with SGs.

Filming conditions for the yeast themselves were not accurately described, and raises an issue of experimental design. Filming conditions and preparations are crucial, as mentioned (but not addressed experimentally) in the introduction that yeast can form stress granules and polar bodies under heat stress. There is no indication that the temperature was being monitored while filming. As for other microscopic preparation, it is simply stated that the cells were “resuspended” after supernatant removal. The conditions in which the cells are filmed should match experimental treatment as closely as possible, in terms of temperature and media. Once again, there is no assessment that demonstrates any true relevance of this phenotype in terms of yeast growth or metabolism. This raises the question as to whether or not the processing body phenotype disassembly documented is an artifact from filming conditions and/or preparation of samples for microscopy. As for quantification and granule counting, no description is given for how the softWoRx analysis algorithm makes its measurements.

The investigators generated plasmids with different partial deletion mutations of Sbp1 in an Sbp1 deletion background. It is observed in supplemental figure 2 that there appears to be an involvement of Sbp1 in stress granule assembly, however this is looked past in favor of the apparent novel disassembly phenotype in supposed P bodies. Figure 4C shows some marginally significant evidence for the AMD mutant and Delta RGG mutant, displaying necessity for the RGG domain. It still leaves the question as to whether the RGG domain is sufficient without the RRM domains.

The author look for involvement of arginine methylation in the RGG repeats of Sbp1 as a means of PB disassembly. They do not directly demonstrate that the arginine in the RGG repeats are actually post-translationally modified in any way. After knocking out methyltransferase Hmt1 and observing no rescue to the disassembly phenotype, the investigators take no additional measure to validate whether other methyltransferases are involved. We also see another granule assembly defect in Hmt1 mutants. They wrongly conclude that the arginine is indeed still being methylated, and by some other residual methyltransferase activity. Roy et al. continue to generate an arginine to alanine substitution mutant. They change the only charged moiety in a low complexity sequence, and do not acknowledge how this could affect the behavior of the protein as a whole. There is also no intermediate number of substitutions to tell if this supposed post translational modification has a cumulative effect on PB disassembly. There is no additional information on how knocking out Hmt1 or substitutions affect yeast growth/metabolism as a whole.

The GST pulldown in figure 5 provides some supporting evidence, however the figure is labeled poorly and makes the gel difficult to interpret. This identifies Edc3 as a potential binding partner, but does not directly implicate the RGG domain as being the region that binds Edc3.

Figure 6 adds an in vitro analysis demonstrating that the addition of RNA and NADH allow Edc3 granules to enlarge. Previous in vitro analysis has been done on *C. elegans* P granules (Saha et al. 2016), and although interesting this has little applicability to an in vivo setting. The addition of Sbp1 decreases the size of the Edc3 granules when there is a 3:1 ratio of Sbp1 to Edc3 in vitro. How the investigators arrived at this ratio is not explained.

Another questionable addition is supplemental figure 4. There is no clear connection between Cuz1 and the rest of the genes of interest.

Overall, I do not feel that the authors have demonstrated a global significance of the phenotype of interest in terms of metabolism and cell growth. They ignore another potentially confounding stress granule phenotype in pursuit of their phenotype of interest. The means of accurately distinguishing stress granules and processing bodies is not adequate, and the gene of interest, Sbp1, seems to be involved with both. Experimental design has significant issues, and conditions to document the phenotype are vaguely explained, leading one to question how well they were monitored. The conclusions made are based on an insufficient amount of appropriate experimental evidence.

Reviewer #2 (Remarks to the Author):

Low complexity RGG-motif sequence is required for Processing body (P-body) disassembly - Roy et al

Previous papers have shown that low-complexity domains are important for driving liquid-liquid phase separation of protein-rich condensates. The RNA granules P-bodies and stress granules are two well studied types of condensates, for which low-complexity domains have been found to be important for their formation. In this study the authors find that the P-body co-localized protein Sbp1 can act as a disassembly factor upon stress relief. They find that specifically the low-complexity domain of Sbp1, composed of an RGG-motif, was necessary for Sbp1 to function as a disassembly factor.

This is an important finding as it identifies a disassembly factor for P-bodies, as well as that low-complexity sequences are important not just for the condensation of RNA granules, but also potentially the disassembly. They also went further and through in vitro work show that Sbp1 directly interacts with Edc3 and can impact Edc3 assemblies in vitro. Overall I believe this manuscript pushes the field forward and recommend the following revisions.

Major comment:

While I understand the historical use from the Parker lab of the CEN based plasmid system to fluorescently labeled RNA granules, this potentially increases the expression of the protein being labeled. As we now know that protein concentration is also an important factor in phase separation, it would be beneficial to fluorescently label Edc3 from the endogenous locus, and verify that the same disassembly defects take place in a $\Delta sbp1$ with wildtype protein expression.

Minor comments:

The writing needs to be cleaned up. For example on line 102 – “Yjef-N domain if Edc3 protein binds” (if should be of). In line 101 PB is used when PBs should be used, and this happened a few other times. Line 125 “device” to devise.

Figure 5A needs better labels for what antibodies are used with what gels.

Figure 6C – panel 4 is labeled “His- -FLAG” when presumably it should be labeled His-Sbp1 Δ RGG.

Figure 6D – panel 2 is missing labels

REVIEWER COMMENTS

Reviewer #1 (Remarks to the Author):

Roy et al. 2021 aims to expand the role of low complexity sequences in the dynamic liquid-liquid phase separation (LLPS) of processing body (PB) granules. The low complexity regions that make up intrinsically disordered regions (IDRs) have been previously implicated in LLPS, and this work could potentially expand on the role of how PB formation and dissolution functions as a dynamic regulator of cellular processes. Investigators attempt to characterize an interesting phenotype in which Sbp1 null mutants form granules under stress conditions that fail to disassemble once placed in recovery conditions. However, enthusiasm for this work is greatly diminished by a general lack of evidence to support the author conclusions. Moreover, there are some significant issues in the experimental design. Based on this, the conclusions of the manuscript are premature.

Author's response:

We have carefully considered all the comments of this reviewer. Importantly we have addressed them experimentally and provided several new pieces of evidence in this revised version which strengthens our conclusion that RGG motif sequences play a critical role in PB disassembly. Two important new experimental results have been briefly described below and also explained in detail in response to the comments made by the reviewer.

A. We provide experimental evidence that Sbp1 binding to Edc3 depends on its RGG-motif (Figure 5A) highlighting the role of RGG-motif in Edc3 granule disassembly. Further, Sbp1 and Edc3 interaction competes with Edc3 self-association (Figure 6) leading to PB disassembly. This result identifies the molecular basis for the role of Sbp1 in disassembly observed in vivo (Figure 1) and in vitro (Figure 7). This result is explained in our response to point#10 of Reviewer#1.

B. We provide experimental evidence that aggregates of EWSR1 (a FET protein implicated in Amyotrophic Lateral Sclerosis) persist longer in absence of Sbp1 protein (Figure 8A & B). Consistent with this observation overexpression of EWSR1 in absence of Sbp1 leads to a growth defect (Figure 8C-E). This result (explained in detail in our response to point#5 of Reviewer#1) indicates that Sbp1 could play a role in disassembly of disease relevant heterologous proteins raising the possibility that low-complexity sequences such as RGG-motif could be used to disassemble pathological aggregates.

For the purpose of simplicity, we have divided the comments of the reviewer in a pointwise manner and addressed each of these points below.

Major issues

1. The authors use transgenic plasmids with reporter genes that label PB and stress

granule (SG) components in two different mutant strains for granule-associated proteins that both contain a low complexity RGG repetitive sequence, Sbp1 and Scd6. They propose the novelty of a low complexity sequence in the disassembly of P bodies seen in their Sbp1 mutant; however they the authors do not accurately distinguish between polar bodies and stress granules. As seen in any figure containing a merge of fluorescently labelled Pab1 and Edc3, there are numerous granules that overlap. Colocalization was not assessed or addressed at any point in the manuscript. This is also made apparent upon the recognition that one gene of interest, Scd6, is present in both PBs and SGs. In figure 2D, an Scd6-mCherry reporter transgene is used to assess the disassembly phenotype in Sbp1 null mutants. There is no discernment between which of the Scd6-labeled granules is a SG or PB, contradicting the conclusions from the first figure in which Sbp1 mutants only show a disassembly for PBs.

Author's response: Edc3 and Pab1 proteins used in our study are conserved and widely-accepted markers of P-bodies and Stress granules, respectively. During recovery conditions in $\Delta sbp1$ background no stress granules are visible (Figure 1A & B). All stress granules disassemble upon 1h of recovery whereas P-bodies clearly persist in $\Delta sbp1$ background. This observation clearly indicates that only P-body disassembly is defective in $\Delta sbp1$ background.

Scd6 is known to localize to both P-bodies and Stress granules (Rajyaguru et al., 2012). As pointed by the reviewer, to confirm if the Scd6-mCherry foci that persist in $\Delta sbp1$ background during recovery are P-bodies, we performed the following experiment. Colocalization experiments using Edc3-mCherry $\Delta sbp1$ strain revealed that majority of Scd6-GFP foci colocalized with Edc3-mCherry (Supp. Figure 2A & B). On the other hand, colocalization studies between Pab1-GFP (stress granule marker) and Scd6-mCherry expectedly did not show any colocalization since the Pab1 foci do not show disassembly defect in absence of Sbp1 (Supp. Figure 2A & B). This indicates that the Scd6-mCherry foci that persist are not stress granules but constitute P-bodies.

2. Sbp1 deletion mutants, as well as some of the other partial deletion constructs also show a defect in SG assembly, however there is no attempt to address or characterize Sbp1 involvement with SGs.

Author's response: We thank the reviewer for this pertinent comment. The SG assembly defect in $\Delta sbp1$ background upon sodium azide stress could be a manifestation of its role as a translation repressor. Several other translation repressors/decapping activators have been shown to affect RNA granule assembly (Beckham et al., 2008; Pilkington and Parker, 2008; Hilliker et al., 2011; Rajyaguru et al., 2012). However the role of Sbp1 is PB disassembly in unexpected and therefore exciting. We therefore focussed on understanding the basis of this observation.

SGs arise from preformed PBs in yeast and therefore it is possible that the P-body disassembly defect and SG assembly defect observed in absence of Sbp1 could be connected. We have highlighted this in 'Discussion' section (Paragraph 4; 343-350).

3. Filming conditions for the yeast themselves were not accurately described, and raises an issue of experimental design. Filming conditions and preparations are crucial, as mentioned (but not addressed experimentally) in the introduction that yeast can form stress granules and polar bodies under heat stress. There is no indication that the temperature was being monitored while filming.

Author's response: We agree with the reviewer that imaging conditions are critical to the outcome of the experiment. In all our experiments we were careful to maintain consistent conditions. All the imaging was performed at room temperature (23-25°C). We have now added this information in the text (Materials and Methods; Microscopy, Para 1).

4. As for other microscopic preparation, it is simply stated that the cells were "resuspended" after supernatant removal. The conditions in which the cells are filmed should match experimental treatment as closely as possible, in terms of temperature and media.

Author's response: Cells were pelleted and resuspended at room temperature in minimal media followed by imaging at room temperature (23-25°C). The temperature conditions were constant for these steps. This is a standard protocol in the field for processing and imaging of yeast cells for looking at P-bodies and stress granules (Nissan and Parker, 2009). We have further elaborated the processing steps in the 'Methods' section.

Overall in response to point#3 and 4, we would like to emphasize that both sample processing and imaging conditions used in our manuscript are standard and well monitored. All the experiments have relevant wild type (BY4741 strain) controls which depict normal assembly and disassembly of RNA granules. Besides, in the same sample/cell where we observe PB-disassembly defect there is no visible SG disassembly defect. These points clearly indicate that the disassembly defect is indeed due to the absence of Sbp1 and not due to cell processing/imaging artifact. This is further confirmed by results of the complementation experiments presented in Figure 4 (and Supp. Figure 5). Expressing Sbp1 on a plasmid in $\Delta sbp1$ strain complements the P-body disassembly defect indicating the requirement of Sbp1 in PB disassembly.

5. Once again, there is no assessment that demonstrates any true relevance of this phenotype in terms of yeast growth or metabolism. This raises the question as to whether or not the processing body phenotype disassembly documented is an artifact from filming conditions and/or preparation of samples for microscopy.

Author response: We have elaborately explained in our response to point#4 (above) that the role of Sbp1 in P-body disassembly is indeed a genuine phenotype. We have now addressed the relevance of the role of low complexity sequences in granule disassembly in disease context. Yeast is an established system for understanding

the pathology of Neurodegenerative disorders such as Amyotrophic lateral sclerosis (ALS), Huntington and Parkinson's disease (Ju et al., 2011; Meriin et al., 2002; Outeiro and Lindquist, 2003).

Ewing's Sarcoma Breakpoint Region protein 1 (EWSR1) is part of FET protein family which has been implicated in neurodegenerative disorders such as Amyotrophic Lateral Sclerosis (ALS). Interestingly these proteins contain low complexity sequences and form pathological aggregates, a phenotype which is recapitulated in yeast. We hypothesized that the role of Sbp1 in disassembly could impact aggregation of above proteins. We have tested this and observe that the EWSR1 aggregates persist in absence of Sbp1 as compared to the wild type strain (Figure 8A & B). Consistent with this observation, overexpression of EWSR1 in the absence of Sbp1 leads to a growth defect (Figure 8C) even though the protein levels of EWSR1 are significantly decreased in absence of Sbp1 (Figure 8D & E). This result highlights that Sbp1 can affect aggregates of disease-relevant heterologous proteins.

6. As for quantification and granule counting, no description is given for how the softWoRx analysis algorithm makes its measurements.

Author's response: We have now provided the citation (Hiraoka et al., 2021) in Methods section which describes the working of SoftWoRx algorithm used by Deltavision microscope for deconvolution of the images. Deconvolution allows imaging at very low light levels which makes multiple focal-plane imaging possible over long periods of time for live light-sensitive specimen. We performed granule counting using ImageJ.

7. The investigators generated plasmids with different partial deletion mutations of Sbp1 in an Sbp1 deletion background. It is observed in supplemental figure 2 that there appears to be an involvement of Sbp1 in stress granule assembly, however this is looked past in favour of the apparent novel disassembly phenotype in supposed P bodies.

Author's response: As explained in detail in our response to point#2, we decided to focus on disassembly defect in absence of Sbp1 since it was unexpected. The role of Sbp1 in Stress granule assembly is likely along the lines of general involvement of translation repressors/decapping activators in RNA granule assembly (Beckham et al., 2008; Pilkington and Parker, 2008; Hilliker et al., 2011; Rajyaguru et al., 2012). However we are grateful to the reviewer for emphasizing the importance of role of Sbp1 in SG assembly. We certainly plan to follow it up in future studies.

8. Figure 4C shows some marginally significant evidence for the AMD mutant and Delta RGG mutant, displaying necessity for the RGG domain. It still leaves the question as to whether the RGG domain is sufficient without the RRM domains.

Author's response: We thank the reviewer for suggesting this experiment. We have tested the sufficiency of Sbp1 RGG-motif in complementing the disassembly defect observed in $\Delta sbp1$ background. This has been presented in Figure 4A & B,

Sbp1 Δ RRM1 Δ RRM2 panel. We observe that the RGG-motif of Sbp1 complements the disassembly phenotype to a similar extent as by the wild type Sbp1. We therefore conclude that the RGG-motif is both necessary and sufficient for P-body disassembly.

9. The author look for involvement of arginine methylation in the RGG repeats of Sbp1 as a means of PB disassembly. They do not directly demonstrate that the arginine in the RGG repeats are actually post-translationally modified in any way. After knocking out methyltransferase Hmt1 and observing no rescue to the disassembly phenotype, the investigators take no additional measure to validate whether other methyltransferases are involved. We also see another granule assembly defect in Hmt1 mutants. They wrongly conclude that the arginine is indeed still being methylated, and by some other residual methyltransferase activity. Roy et al. continue to generate an arginine to alanine substitution mutant. They change the only charged moiety in a low complexity sequence, and do not acknowledge how this could affect the behavior of the protein as a whole. There is also no intermediate number of substitutions to tell if this supposed post translational modification has a cumulative effect on PB disassembly. There is no additional information on how knocking out Hmt1 or substitutions affect yeast growth/metabolism as a whole.

Author's response: We would like to politely emphasize that we have not concluded in the manuscript about the role of Sbp1 methylation in disassembly. We have mentioned in 'Discussion' of the manuscript that the role of Sbp1 methylation in disassembly is a possibility (paragraph 3; line 319-329). We mention this since the arginine methylation defective (AMD) mutant (like the RGG-motif deletion mutant) clearly fails to rescue the disassembly defect in absence of Sbp1 (Supp. Figure 5A).

In response to the queries raised by the reviewer about methylation status of Sbp1, we would like to draw the attention to the point that arginine methylation of Sbp1 has been reported (Bhatter et al., 2019). The arginine methylation defective (AMD) mutant of Sbp1 has also been previously characterized (Bhatter et al., 2019). A similar AMD mutant has been reported for Scd6 as well (Poornima et al., 2016). The arginine to alanine substitutions do not affect the folding and stability of the Sbp1 mutant based on a) protein level studies, b) CD analysis, and c) RNA binding analysis (Figures 2D, H & J in Bhatter et al., 2019). We mutated all the RGG-motif arginine to alanine as mutations of fewer arginine leads to partial phenotypes in our previous studies suggesting a dosage effect (Bhatter et al., 2019; Poornima et al., 2016). The AMD substitution mutant has been characterized for its effect on yeast growth metabolism. It rescues the overexpression growth defect phenotype observed with wild type Sbp1 (Figure 2G; Bhatter et al., 2019). We agree that the results using AMD do not conclusively prove a causative role of arginine methylation in disassembly but provide a good starting point for detailed analysis in future studies. As mentioned in the manuscript, the role of a backup methyltransferases (if any) in Sbp1-mediated disassembly will be interesting. We have looked at possible role of Rmt2 (another important yeast methyltransferase) as a backup

to Hmt1 and observed that it does not affect Sbp1 function (Bhatter et al., 2019). Unlike in mammals, arginine MTases other than Hmt1 and Rmt2 are very poorly characterized in yeast. Addressing their role in general and in context of granule disassembly is an exciting future direction because the role of arginine methylation in disassembly is not the focus of our current study.

10. The GST pulldown in figure 5 provides some supporting evidence, however the figure is labelled poorly and makes the gel difficult to interpret. This identifies Edc3 as a potential binding partner, but does not directly implicate the RGG domain as being the region that binds Edc3.

Author's response: We have now tested the role of RGG-deletion mutant of Sbp1 in binding Edc3 using recombinant purified proteins. We observe that the RGG-deletion mutant of Sbp1 is defective in binding to Edc3. This is now presented as Figure 5A. This data proves the role of RGG-motif of Sbp1 in binding Edc3. The data previously presented as 5A & B is now shown as 5B and C.

Importantly, we have added another experimental evidence that provides mechanistic insight into the direct role of Sbp1 in Edc3 disassembly. It is well-known that Edc3 self-association mediated by its C-terminal low complexity YjeF-N domain is important for P-body assembly (Decker et al., 2007). We hypothesized that Sbp1-Edc3 interaction could compete with Edc3 self-association to orchestrate PB disassembly. Results obtained using purified recombinant proteins indicate that Edc3-Sbp1 interaction competes with the interaction between the YjeF-N domain and full-length Edc3 (Figure 6) thereby providing further insight into the mechanism of PB disassembly by Sbp1. This results is consistent with the data presented in Figure 7 where we observe that purified Sbp1 disrupts Edc3 assemblies.

11. Figure 6 adds an in vitro analysis demonstrating that the addition of RNA and NADH allow Edc3 granules to enlarge. Previous in vitro analysis has been done on C. elegans P granules (Saha et al. 2016), and although interesting this has little applicability to an in vivo setting. The addition of Sbp1 decreases the size of the Edc3 granules when there is a 3:1 ratio of Sbp1 to Edc3 in vitro. How the investigators arrived at this ratio is not explained.

Author's response: Phase separation of proteins is being increasingly appreciated to contribute to formation of higher order assemblies such as RNA granules. Demonstrating the phase separation behaviour of Edc3 is not the purpose of Figure 7 (earlier Figure 6). It is to address the direct role of Sbp1 in phase separation behaviour of Edc3. The result presented in Figure 7 provides mechanistic basis of a direct role of Sbp1 in P-body disassembly. The new result (Figure 6A & B) pertaining to competition between Edc3-Sbp1 and Edc3-Edc3 interaction provide further mechanistic insight into the results presented in Figure 7 (*in vitro*) and Figure 1 (*in vivo*).

However Figure 7 also provided critical data about the direct role of RNA and NADH in formation of Edc3 assemblies. A previous report suggested a possible yeast Edc3-NADH binding (Walters et al., 2014). It was proposed that such interaction could affect Edc3 function however the basis for this was unclear. Our data provides evidence for how NADH could modulate Edc3 function by directly affecting its phase separation behaviour (Figure 7, NADH panel).

Regarding usage of 30uM of Edc3 protein, we sincerely apologize for the confusion that arose due to inaccurate writing on our part. All proteins in Figure 7 have been used at 10uM concentration. This was correctly mentioned in the Methods section however in the legends to Figure 7 we have inadvertently mentioned that 30 uM (instead of 10 uM) of Edc3 was used. This mistake has been rectified. We thank the reviewer for pointing this out.

12. Another questionable addition is supplemental figure 4. There is no clear connection between Cuz1 and the rest of the genes of interest.

Author's response: ZFAND1 is the human homolog of Cuz1. The rationale of testing the role of Cuz1 was to check if the Stress granule disassembly mechanism mediated by ZFAND1 in HEK293T cells (Turakhiya et al., 2018) holds true for stress granules and P-bodies in yeast. We agree that Cuz1 is not related to Scd6 and Sbp1. We can exclude this data (now presented as Supp. Figure 7) if the reviewer and editor think that it does not help the narrative.

13. Overall, I do not feel that the authors have demonstrated a global significance of the phenotype of interest in terms of metabolism and cell growth. They ignore another potentially confounding stress granule phenotype in pursuit of their phenotype of interest. The means of accurately distinguishing stress granules and processing bodies is not adequate, and the gene of interest, Sbp1, seems to be involved with both.

Author's response: We have addressed experimentally the concerns of the reviewer raised in all previous points. We have also experimentally addressed the only concern raised by the other reviewer (see our response below). Additionally we provide two new results that impart mechanistic insight and highlight physiological relevance of the results presented in this manuscript.

Competition between Sbp1-Edc3 and Edc3-Edc3 interaction (Figure 6) provides the basis for the role of Sbp1 in disassembly in vivo (Figure 1) and in vitro (Figure 7).

Persistence of aggregates of FET protein EWSR1 in the absence of Sbp1 (Figure 8A & B) highlights that the role of low complexity sequences in granule disassembly is much broader and applicable to heterologous disease-relevant proteins. Additionally overexpression of EWSR1 in $\Delta sbp1$ background leads to a growth defect (Figure 8C). We believe that this result will provide an important starting point to researchers in the field to explore the role of low complexity sequences in modulating pathological protein aggregates observed in different diseases.

The concern regarding distinguishing P-bodies and Stress granules has been addressed (please see our response to point#1 with new experimental data).

We agree with the reviewer that Sbp1 is involved in assembly of Stress granules and disassembly of P-bodies. As explained in our response to point#2 & 7, in this study we have focussed on P-body disassembly phenotype of Sbp1 since the role of RGG-motif in disassembly is unexpected.

14. Experimental design has significant issues, and conditions to document the phenotype are vaguely explained, leading one to question how well they were monitored. The conclusions made are based on an insufficient amount of appropriate experimental evidence.

Author's response: As mentioned elaborately in our response to point#3, 4, 5, and 6 we have followed standard protocols for growing, processing and imaging live cells. We have added details of processing and imaging of cells in the 'Methods' section wherever applicable to provide more clarity. We have addressed experimentally the concerns raised by the reviewer. Additionally we have provided two important experimental results highlighting the mechanistic basis of Sbp1 role in disassembly and relevance of this role in disease context.

Reviewer #2 (Remarks to the Author):

Low complexity RGG-motif sequence is required for Processing body (P-body) disassembly - Roy et al

Previous papers have shown that low-complexity domains are important for driving liquid-liquid phase separation of protein-rich condensates. The RNA granules P-bodies and stress granules are two well studied types of condensates, for which low-complexity domains have been found to be important for their formation. In this study the authors find that the P-body co-localized protein Sbp1 can act as a disassembly factor upon stress relief. They find that specifically the low-complexity domain of Sbp1, composed of an RGG-motif, was necessary for Sbp1 to function as a disassembly factor.

This is an important finding as it identifies a disassembly factor for P-bodies, as well as that low-complexity sequences are important not just for the condensation of RNA granules, but also potentially the disassembly. They also went further and through in vitro work show that Sbp1 directly interacts with Edc3 and can impact Edc3 assemblies in vitro. Overall I believe this manuscript pushes the field forward and recommend the following revisions.

Author's response: We thank the reviewer for his/her encouraging comments.

Major comment:

While I understand the historical use from the Parker lab of the CEN based plasmid

system to fluorescently labelled RNA granules, this potentially increases the expression of the protein being labelled. As we now know that protein concentration is also an important factor in phase separation, it would be beneficial to fluorescently label Edc3 from the endogenous locus, and verify that the same disassembly defects take place in a $\Delta sbp1$ with wildtype protein expression.

Author's response: We thank the reviewer for this insightful comment. We have created $\Delta sbp1$ deletion in the background of endogenously tagged Edc3-mCherry strain. We observe that the disassembly of Edc3-mCherry is defective in the absence of Sbp1 (Figure 1D & E). This result confirms that the disassembly defect observed with plasmid-encoded Edc3-mCherry is not due to overexpression of Edc3.

Minor comments:

The writing needs to be cleaned up. For example on line 102 – “Yjef-N domain if Edc3 protein binds” (if should be of). In line 101 PB is used when PBs should be used, and this happened a few other times. Line 125 “device” to devise.

Author's response: We have corrected the mistakes.

Figure 5A needs better labels for what antibodies are used with what gels.

Author's response: Figure 5A is now presented as Figure 5B. We have labelled the Figure 5B appropriately to easily understand the data.

Figure 6C – panel 4 is labelled “His- -FLAG” when presumably it should be labelled His-Sbp1 Δ RGG.

Author's response: Figure 6C is now presented as Figure 7C. We have corrected the labelling.

Figure 6D – panel 2 is missing labels

Author's response: Figure 6D is now presented as Figure 7D. We have added the labels.

We have gone through the manuscript again and corrected a few other mistakes as well. All the changes have been marked.

REVIEWERS' COMMENTS

Reviewer #1 (Remarks to the Author):

I'm satisfied with the authors additions to the manuscript.

Reviewer #2 (Remarks to the Author):

I am happy with the revisions made by the author and am okay with publication of this manuscript in Nature Communications.

If the authors were up to further address Reviewer 1's concerns that the granules they are seeing after recovery are indeed bona fide P-bodies, they could potentially use another P-body marker like Dcp2 which is generally considered more a core P-body component like Edc3 versus something like Dhh1 which is found in both P-bodies and stress granules. They even list Dcp2-mCherry in their plasmids, but I could not find its use in the manuscript.

Response to reviewer's comments:

Reviewer #1 (Remarks to the Author):

I'm satisfied with the authors additions to the manuscript.

Author's response: We thank the reviewer for providing many insightful suggestions to help us improve this manuscript.

Reviewer #2 (Remarks to the Author):

I am happy with the revisions made by the author and am okay with publication of this manuscript in Nature Communications.

If the authors were up to further address Reviewer 1's concerns that the granules they are seeing after recovery are indeed bona fide P-bodies, they could potentially use another P-body marker like Dcp2 which is generally considered more a core P-body component like Edc3 versus something like Dhh1 which is found in both P-bodies and stress granules. They even list Dcp2-mCherry in their plasmids, but I could not find its use in the manuscript.

Author's response: We thank the reviewer for this suggestion. We aim to do this experiment using several other markers in future as part of our endeavour to identify and characterise the protein and RNA components of the disassembly-defective granules. This would be part of another manuscript.

However, we believe that the results presented in the manuscript, sufficiently indicate that the granules that are defective in disassembly represent P-bodies (PB). We have learned this based on the use of three conserved markers (Edc3, Scd6 and Dhh1). Another observation that contributes to this understanding is that stress granule (SG) disassembly is not defective under the conditions where P body disassembly is defective. Since PB and SG are the two main kinds of RNA granules in yeast, we conclude that granules containing Edc3, Scd6 and Dhh1 that are visible during recovery phase are indeed represent P-bodies.